# Assessment of stochastic weather forecast of precipitation near European cities, based on analogs of circulation

Meriem Krouma [1,2], Pascal Yiou [2], Céline Déandreis [1], and Soulivanh Thao [2]

[1]ARIA Technologies, 8 Rue de la Ferme, 92100 Boulogne-Billancourt, France
[2]Laboratoire des Sciences du Climat et de l'Environnement, UMR 8212 CEA-CNRS-UVSQ, IPSL & Université Paris-Saclay, 91191 Gif-sur-Yvette, France

**Correspondence:** Meriem Krouma (meriem.krouma@lsce.ipsl.fr)

**Abstract.**

In this study, we aim to assess the skill of a stochastic weather generator (SWG) to forecast precipitation in several cities of Western Europe. The SWG is based on random sampling of analogs of the geopotential height at 500 hPa (Z500). The SWG is evaluated for two reanalyses (NCEP and ERA5). We simulate 100-member ensemble forecasts on a daily time increment. We evaluate the performance of SWG with forecast skill scores and we compare it to ECMWF forecasts.

Results show significant positive skill score (continuous rank probability skill score and correlation) comparing to persistence and climatology forecasts for lead times of 5 and 10 days for different areas in Europe. We find that the low predictability of our model is related to specific weather regimes, depending on the European region. Comparing SWG forecasts to ECMWF forecasts, we find that the SWG shows a good performance for 5 days. This performance varies from one region to another. This paper is a proof of concept for a stochastic regional ensemble precipitation forecast. Its parameters (e.g. region for analogs) must be tuned for each region in order to optimize its performance.

## 1 Introduction

Ensemble weather forecasts were designed to overcome the issues of meteorological chaos, from which small uncertainties in initial conditions can lead to a wide range of possible trajectories (Sivillo et al., 1997; Palmer, 2000). Hence, from a sufficiently large ensemble of initial conditions, it is in principle possible to sample the probability distribution of future states of the system.

Forecasts issued by meteorological centers are obtained by computing several simulations with perturbed initial conditions, in order to sample uncertainties. Those experiments are rather costly in terms of computing resources and are generally limited to a few tens of members (Hersbach et al., 2020; Toth and Kalnay, 1997), which can hinder a proper estimate of probability distributions of trajectories. Moreover, obtaining information at local spatial scales can be difficult because the horizontal resolution of the atmospheric models is around 18 km, e.g. for the European Centre for Medium-Range Weather Forecasts (ECMWF) ensemble forecast system.

From a mathematical point of view, computing the probability distribution of the trajectories of a (deterministic) system makes the underlying assumption that the system behaves like a stochastic process, for which statistical properties are defined naturally (Ruelle, 1979; Eckmann and Ruelle, 1985). This has justified the development of stochastic weather generators

(SWG), which are stochastic processes that emulate the behavior of key climate variables (Ailliot et al., 2015). The advantages of stochastic models are a relative simplicity of implementation and a low computing cost. The challenge of their development is to verify that the behavior of the simulations are realistic, according to well-defined criteria (van den Dool, 2007; Jolliffe and Stephenson, 2011).

The first stochastic weather generators were devised to simulate rainfall occurrence by Gabriel and Neumann (1962) and to simulate rainfall amounts by Todorovic and Woolhiser (1975). SWGs were developed and used to estimate the probability distributions of climate variables such as temperature, solar radiation, and precipitation through extensive simulations (Richardson, 1981).

Stochastic weather generators can be useful complements to atmospheric circulation models, in order to simulate large ensembles of local variables, as they can be calibrated for small spatial scales comparing to numerical models (Ailliot et al., 2015). This explains their wide applications in impact studies.

A successful simulation with SWG relies on the choice of inputs. One of them consists in the use of the atmospheric circulation as a predictor for other local variables. The (loose) rationale for this choice is that the circulation is modeled by prognostic equations (Peixoto and Oort, 1992), that drive the other physical variables. Therefore the primitive equations of the atmosphere (Peixoto and Oort, 1992, Chap. 3) suggest that reproducing temporal variability on daily time scales requires considering circulation variables. The influence of large-scale circulation on local climate variables has been proven in previous studies such as the influence of atmospheric circulation on Mediterranean Basin (Mastrantonas et al., 2021) and Greece precipitation (Xoplaki et al., 2000; Türkes et al., 2002). Similar influences have been found on precipitation and temperature over the North Atlantic region (Jézéquel et al., 2018b).

Analogs of circulation were initially designed to provide "model-free" forecasts, by assuming that similar situations in atmospheric circulation may lead to similar local weather conditions (Lorenz, 1969). The potential to simulate large ensembles of forecasts temperature with circulation analogs was explored by Yiou and Déandréis (2019), by considering random resamplings of $K$ best analogs (rather than only considering the best analog). This has lead to the development of a SWG in "predictive" mode, which uses updates of reanalysis datasets (Kistler et al., 2001) as input.

Alternative systems of analogs to forecast precipitation have been proposed by Atencia and Zawadzki (2014). Those systems are based on analogs of precipitation itself. Such systems are very efficient for nowcasting, i.e. forecasting precipitation within the next few hours. Considering the atmospheric circulation analogs allows to focus on longer time scales.

Yiou and Déandréis (2019) evaluated ensemble forecasts of the analog SWG for temperature and the NAO index with classical probability scores against climatology and persistence. Reasonable scores were obtained up to 20 days. Through this study, we aim to assess the skill of this SWG to forecast precipitation in different areas of Europe and for different lead times. The previous study on this forecast tool was a proof of concept for temperature. In this study, we will adapt the parameters of the analog SWG to optimize the simulation of European precipitations. We then analyse the performance of this SWG for lead times of 5 to 20 days, with the forecast skill scores used by Yiou and Déandréis (2019).

We will evaluate the seasonal dependence of the forecast skills of precipitation and the conditional dependence to weather regimes. Finally, comparisons with medium range precipitation forecasts from the ECMWF will be performed.

The paper is divided as follows: Section 2 is dedicated to describe the data used for the experiments. Section 3 explains the methodology (analogs, stochastic weather generator and forecast skill scores). Section 4 details the experimental set up and justifies the choices that we made for the forecast parameters. Section 5 details results of simulations and the evaluation of the ensemble forecast. Section 6 contains the main conclusions of the analyses.

## 2  Data

Daily precipitation data were obtained from the European Climate Assessment and Data (ECAD) project (Klein Tank et al., 2002) for four locations in western Europe (Berlin, Madrid, Orly, Toulouse), which are subject to contrasted meteorological influences (Figure 1). ECAD provides station data, that are available at a daily time step from 1948 to 2019. The choice of those stations was based on the availability of large and common period of observations with a low rate of missing data (less than 10%). For verification issues, we used also the E-Obs data (Haylock et al., 2008), which are a daily gridded data available from 1979 to present with a horizontal resolution of $0.25° \times 0.25°$. E-Obs data are spatial interpolations of ECAD data.

We recovered the geopotential height at 500 hPa (Z500) and sea level pressure (SLP) fields from the reanalysis of the National Centers for Environmental Prediction (NCEP: Kistler et al. (2001)) with a spatial resolution of $2.5° \times 2.5°$ from 1 January 1948 to 31 December 2019.

We also used the atmospheric reanalysis (version 5) of the European Centre for Medium-Range Weather Forecasts (ECMWF) (ERA5; Hersbach et al. (2020)). ERA5 data are available from 1950 to present with a horizontal resolution of $0.25° \times 0.25°$. There are fundamental differences between the two reanalyses, in the atmospheric models, assimilated data, and assimilation schemes.

We considered the daily averages of Z500 from NCEP and ERA5, over the region covering 30°W – 20°E and 40°– 60°N to compute circulation analogs. Daily averages of SLP were used over the region covering 80°W – 20°E and 30° – 70°N to define weather regimes.

In order to assess the predictive skill of our precipitation forecast model, a comparison with another forecast was made. There are many available datasets that can be used for deriving this information. We considered the ECMWF ensemble forecast dataset system 5 (Vitart et al., 2017). It is a daily gridded dataset interpolated over Europe to provide information covering the all the domain. Data are available through the Copernicus Climate Data Store including forecasts created in real-time (since 2017) and hindcast forecasts from 1993 to 2019 (Vitart et al., 2017). The data are provided at an hourly time step with a horizontal resolution of $0.25° \times 0.25°$. We considered the grid points that include Berlin, Orly, Toulouse and Madrid, which were identified in the ECAD database.

## 3 Methodology

### 3.1 Analogs

The first step is to build a database of analogs of the atmospheric circulation. We outline the procedure of Yiou and Déandréis (2019), summarized in Figure 1a. For a given day $t$, we determine the similarity of Z500 for all days $t'$ that are within 30 calendar days of $t$ but in a different year from $t$. The similarity is quantified by a Euclidean distance (or root mean square) between the daily Z500 maps. Other types of distances are possible (Blanchet et al., 2018), but the expected impact on the results is often marginal (Toth, 1991). We believe that the simplicity of the a Euclidean distance makes it more robust to changes in horizontal resolution (e.g. from NCEP to ERA5), compared to more sophisticated distances that include local spatial gradients, which would require adjustments and additional tuning. This choice can be left open for future fine tuning, depending on the region.

For each day $t$, we consider the $K$ best analogs, i.e. for which the distances are the smallest. We compute the spatial rank correlation between the Z500 best analogs and the Z500 at time $t$ for a posteriori verification purposes.

As a refinement over the study of Yiou and Déandréis (2019), a time embedding of $\tau$ days was used for the search of analogs dates. This means that the field $X(t)$ for which we compute analogs is $X(t) = (Z500(t), Z500(t+1), \ldots, Z500(t+\tau))$. This ensures that temporal derivatives of the atmospheric field are preserved (Yiou et al., 2013). Hence the distance that is optimized to find analogs of the $Z500(x,t)$ field is:

$$D(t,t') = \left[ \sum_x \left( \sum_{i=0}^{\tau} |Z500(x,t+i) - Z500(x,t'+i)|^2 \right) \right]^{\frac{1}{2}}, \tag{1}$$

where $x$ is a spatial index, $\tau$ is the embedding time.

We consider different geographic domains as showed in Figure 1 for the computation of analogs and weather regimes. The computation of circulation analogs was performed with the "blackswan" Web Processing Service (WPS, Hempelmann et al. (2018)). The "blackswan" WPS is an online tool that helps computing circulation analogs on various datasets (reanalyses, climate model simulations) with a user friendly interface.

### 3.2 Configuration of stochastic weather generator

We use a stochastic weather generator (SWG) based on a random sampling of the circulation analogs. The operation of the SWG and its design are detailed by Yiou and Déandréis (2019). The aim is to generate random trajectories from the previously computed analogs. Therefore, to generate a trajectory, we proceed as follows. For a given day $t_0$ in year $y_0$, we generate a set of $N = 100$ simulations until a time $t_0 + T$, with a lead time $T \in \{5, 10, 20\}$ days. We start at day $t_0$ and randomly select an analog (out of $K$ analogs) of day $t_0 + 1$. The random selection of analogs of day $t_0 + 1$ is performed with weights that are proportional to the calendar difference between $t_0$ and analog dates, to ensure that time goes forward. We also exclude analog dates with years that are equal to $y_0$. This rule is important for the next iterations. We then replace $t_0$ by the selected analog of

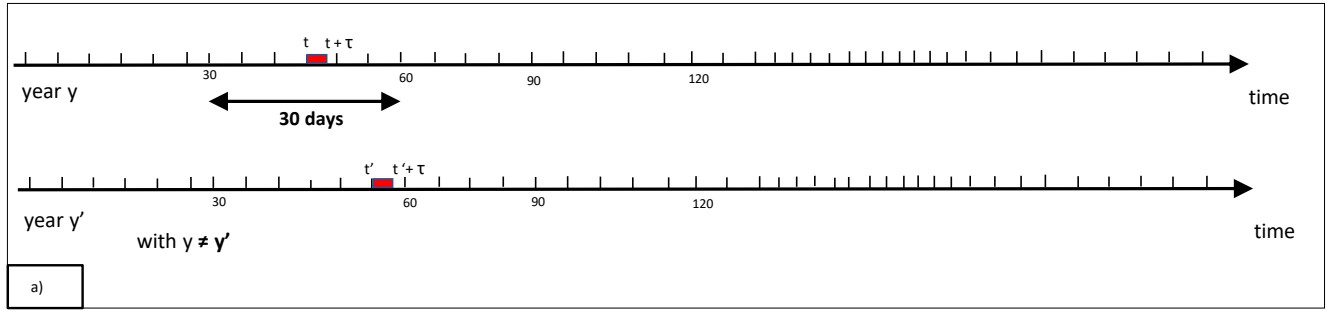

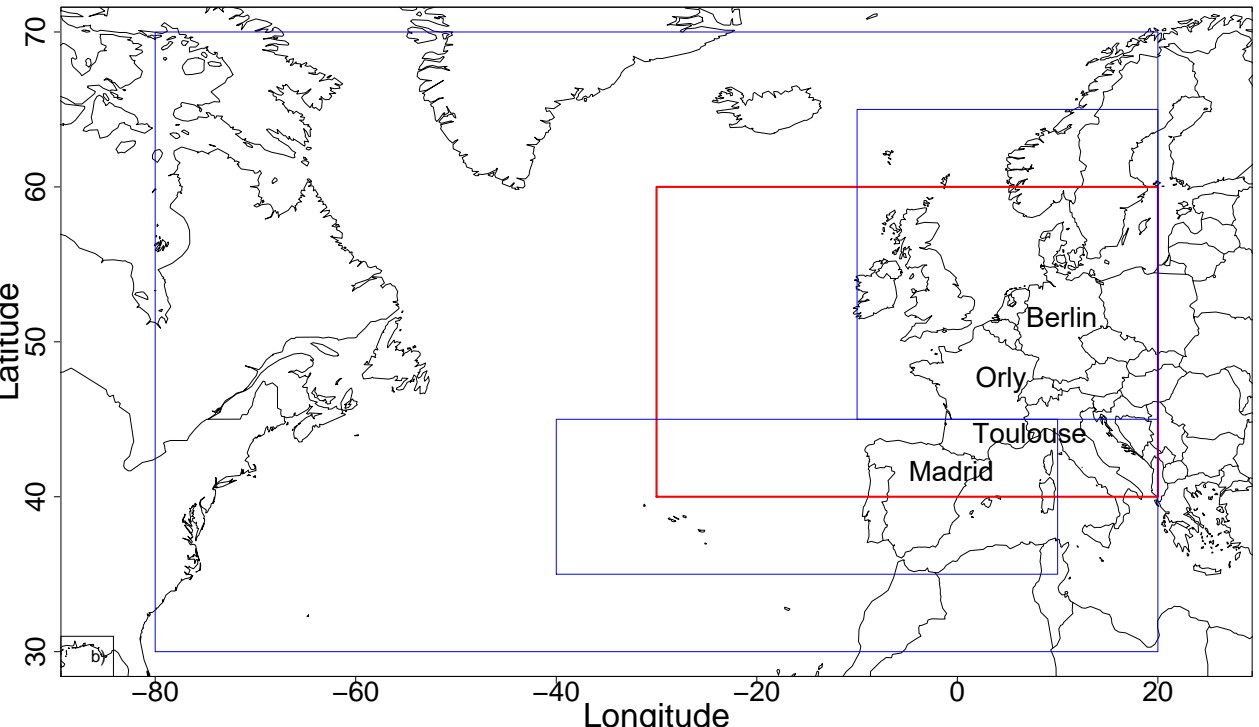

**Figure 1.** Parameters of the analog computation. (a) For each day $t$ in year $y$, we chose an analog day $t'$ with a similar sequence of $\tau$ consecutive day Z500 patterns. $t'$ is selected within 30 calendar days of $t$, and in a year $y' \neq y$. (b) Domains of computation of analogs, we computed analogs over different domains, each one includes a part of the Atlantic and focus in a part of Western Europe, in order to test the sensitivity of our model to different geographic areas, the optimising area was [30°W-20°E; 40°-60°N], indicated by the red rectangle.

$t_0 + 1$ and repeat the operation $T$ times. Excluding analog selection in year $y_0$ ensures that we do not use information from the $T$ days that follow $t_0$. Hence we obtain a hindcast trajectory between $t_0$ and $t_0 + T$.

This operation of trajectory simulation from $t_0$ to $t_0 + T$ is repeated $N = 100$ times. The daily precipitation of each trajectory is time-averaged between $t_0$ and $t0 + T$. Hence, we obtain an ensemble of $N = 100$ forecasts of the average precipitation for day $t_0$ and lead time $T$.

Then $t_0$ is shifted by $\Delta t \geq 1$ days, and the ensemble simulation procedure is repeated. This provides a set of ensemble forecasts with analogs.

We made a hindcast exercise where the forecasts of precipitations based on atmospheric circulation (Z500) are started every $\Delta t \approx T/2$ day between January 1, 1948 and December 31, 2019. This yields a stochastic ensemble hindcast of precipitation and atmospheric circulation (Z500). In this paper, we therefore analyze the properties of an ensemble forecast of mean precipitation between $t_0$ and $t_0 + T$. To evaluate our forecasts, the predictions made with the SWG are compared to the persistence and climatological forecasts. The persistence forecast consists of using the average value between $t_0 - T$ and $t_0$ for a given year. The

climatological forecast takes the climatological mean between $t_0$ and $t_0 + T$. The two "reference" forecasts are randomized by adding a small Gaussian noise, whose standard deviation is estimated by bootstrapping over $T$ long intervals. We thus generate sets of persistence forecasts and climatological forecasts that are consistent with the observations (Yiou and Déandréis, 2019) .

The simulations of this stochastic model will be called "SWG forecasts", as opposed to ECMWF forecasts.

### 3.3    Forecast Verification

Forecast verification is the process of determining the statistical quality of forecasts. A wide variety of ensemble forecast verification procedures exists. They involve measures of the relationship between a set of forecasts and corresponding observations. To assess the quality of precipitation forecasts, we compute indicators such as the Correlation and Continuous Rank Probability Skill Score (CRPSS) for each lead time $T$, for different seasons and months.

The temporal rank correlation (referred as correlation skill) is calculated between the precipitation observations and the

median of 100 simulations. This simple diagnostic is often used to assess forecast skills of indices (Scaife et al., 2014).

The continuous ranked probability score (CRPS) is widely used for probability forecast verifications (Ferro, 2007). It is sensitive to the distance between forecast and observation probability distributions.

If the ensemble forecast $x$ yields a probability distribution $P(x)$ for a value $x_a$, the CRPS measures how the probability distribution of $x$ compares with $x_a$ (Hersbach, 2000).

The CRPS is computed as:

$$CRPS(P, x_a) = \int\limits_{-\infty}^{+\infty} \left( P(x) - \mathcal{H}(x - x_a) \right)^2 dx, \tag{2}$$

where $x_a$ is the observation and $\mathcal{H}$ is the Heaviside function of the occurrence of $x_a$ ($\mathcal{H}(y) = 1$ if $y \geq 0$, and $\mathcal{H}(y) = 0$ otherwise). The decomposition and properties of the CRPS have been investigated by Ferro (2007), Hersbach (2000), and Zamo and Naveau (2018). A perfect forecast would have a CRPS equal to 0, but the CRPS value obviously depend on the units

of the variable to forecast, so that quantifying what is a "good" forecast requires a normalization. It is hence difficult to compare

CRPS values for temperature and precipitation, within the same ensemble forecast. This issue is also acute for non Gaussian variables with heavy tails (Zamo and Naveau, 2018), so that the interpretation of a given CRPS value might not always be informative.

One way of circumventing this difficulty is to compare CRPS values to reference forecasts, such as persistence or climatology. The continuous rank probability skill score (CRPSS) is a normalization of Eq. (2) with respect to such a reference.

The CRPSS is hence computed by:

$$CRPSS = 1 - \frac{\overline{CRPS}}{\overline{CRPS_{ref}}} \tag{3}$$

where $\overline{CRPS}$ is the time average of the $CRPS$ of the SWG forecast and $\overline{CRPS_{ref}}$ is the time average of the $CRPS$ of the reference (either climatology or persistence). The CRPSS is interpreted as a fraction of improvement over a reference forecast.

The values of the CRPSS vary between $-\infty$ and 1. The forecast is considered to be an improvement over the reference when the CRPSS value is close to 1 (i.e. when the CRPS is 0). Values of CRPSS equal to 0 indicate no improvement over the reference. Values inferior to 0 mean that the forecast is worse than the reference.

We use the CRPSS values to determine the maximum lead time $T$ for which the SWG forecast is better than references. Then the SWG assessments will use the CRPS and directly compare the probability distributions of precipitation ensemble forecasts.

### 3.4 Dependence of forecast on weather regimes

We investigate the role of North Atlantic weather patterns on the forecast quality by attributing CRPS values of the SWG precipitation simulations to weather regimes. Weather regimes are defined as large-scale quasi stationary atmospheric states. They are characterised by their recurrence, persistence and stationarity (Michelangeli et al., 1995). They help describing the features of the atmospheric circulation. Surface variables like temperature and precipitation are largely correlated with weather regimes (van der Wiel et al., 2019) .

The North Atlantic weather regimes were computed with the procedure of Yiou et al. (2008), with the NCEP reanalysis. The first 10 principal components of SLP (large region in Figure 1b) are classified with a k-means algorithm onto four classes, over a reference period between 1970 and 2010. The procedure is repeated 100 times with random k-means initialization. Then we classify the resulting $100 \times 4$ k-means weather regimes, in order to determine the most probable classification. This heuristic procedure increases the robustness of the obtained weather regimes. Figure 2 shows four weather regimes for each season (winter and summer) that are coherent with the literature (Cassou et al., 2011; Ghil et al., 2008; Kimoto, 2001; Michelangeli et al., 1995)

The winter weather regimes are the Scandinavian blocking (BLO), Atlantic ridge (AR), negative phase of the North Atlantic oscillation (NAO-) and Zonal flow (ZO). The summer weather regimes are the negative phase of the NAO (NAO-), Atlantic ridge (AR), Scandinavian blocking (BLO) and Atlantic low (AL). The regimes are not the same in both seasons, due to the seasonality of the large scale atmospheric circulation.

For each day (in winter and summer) between 1948 and 2019, we classify the SLP by minimizing the root mean square to four reference (1970–2010) weather regimes.

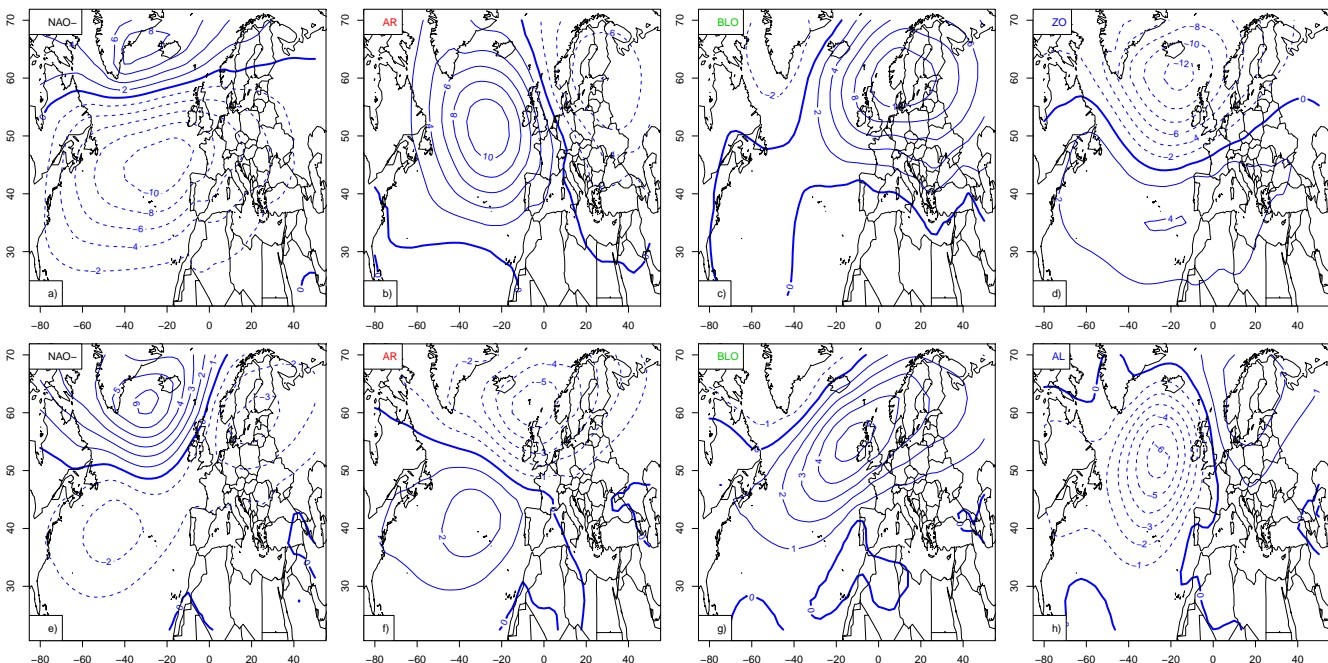

**Figure 2.** Weather regimes over Europe from SLP fields: North Atlantic oscillation (NAO-),the Atlantic ridge (AR), the Scandinavian blocking (BLO), and Atlantic zonal (NAO+). The figure summarises the different states of the atmosphere during summer (a to d) and winter (e to h). It indicates the low and the high pressure over Europe and the direction of flow from the west (Atlantic) to the east. The isolines show seasonal anomalies with respect to a June-July-August and December-January-February means, in hPa with 2 hPa increments.

For each day $t$ (within a given season), we consider the analogs dates of all $N = 100$ simulations between $t$ and $t + T$ and the corresponding classification into weather regimes. Then we determine the most frequent weather regime of the $N$ member ensemble forecast between $t$ and $t + T$. We hence obtain times series on the most likely weather pattern that dominates in the ensemble forecast between $t$ and $t + T$.

We evaluate the influence of the dominating weather regimes on the SWG forecast quality by plotting the probability distribution of CRPS values *conditional* to each weather regime. This is done separately for "good" forecasts (low CRPS values) and "poor" forecasts (high CRPS values).

We identify two classes of predictability from CRPS values:

– Low predictability is related to high values of CRPS that exceed the 75th quantile,

– High predictability is linked to low values of CRPS, below the 25th quantile.

Then we associate the dominating weather regimes computed above with classes of high or low predictability. This procedure helps identifying atmospheric patterns that could lead to low or high predictability with the SWG model.

## 4 SWG parameter optimization

In order to obtain good forecast skill, we start by verifying the relationship between Z500 over the Euro-Atlantic region and the precipitation in the four studied areas. We show the maps of the temporal rank correlation between the daily average of
200 Z500 and the precipitation in Appendix B1. We found significant negative correlation between Z500 and the precipitation with p-values $\leq 0.05$.

Then, we adjust the parameters of the SWG simulations to obtain better forecasts. The first parameter is the geographical area. We computed sample trajectories of the SWG for the four domains outlined in Figure 1b. We used different domains in order to find an optimal region which allows verifying the relationship between precipitation and Z500 for the four studied
areas. Each domain includes a part of the Atlantic and a part of western Europe. We choose a widest domain with the coordinates 80°W – 20°E and 30°– 70°N in order to catch the variability in the whole Euro-Atlantic region. However, it gave poor skill scores for precipitation forecasting for the studied areas as shown in Table 1. Then we focused on two smaller domains (outlined in blue in Figure 1b): one centred over northern Europe and the other centred over southern Europe. We found good forecast skills for specific locations. Same level of performance was found for the domain (outlined in red in Figure 1b) with
coordinates 30°W – 20°E and 40° – 60°N. Therefore, we kept this domain for the subsequent analyses, because it allows to optimise the correlations between Z500 and precipitation for the four studied areas and the time of computation of analogs at the same time. We compared the skill scores over the geographic domain with the coordinates 80°W – 20°E ; 30° – 70°N and 30°W – 20°E ; 40° – 60°N. We determined that the SWG simulations showed better skill for the geographic domain (outlined in red in Figure 1b) and the skill scores remained the highest ones as represented in the following Table 1.

**Table 1.** Correlation between observations and the median of 100 simulations for the winter (DJF) for the different studied domains represented in Figure 1b, with the coordinates 80°W – 20°E ; 30° – 70°N for the largest one (blue) and 30°W – 20°E ; 40° – 60°N for the red rectangle for a lead time of 5 days.

| Location | [80°W – 20°E ; 30° – 70°N] domain | | [30°W – 20°E ; 40° – 60°N] domain | |
|---|---|---|---|---|
| | Correlation | 95% confidence interval | Correlation | 95% confidence interval |
| **Berlin** | 0.32 | 0.30 – 0. 35 | 0.50 | 0.48 – 0.56 |
| **Madrid** | 0.35 | 0.33 – 0. 39 | 0.53 | 0.51 – 0.55 |
| **Orly** | 0.39 | 0.37 – 0. 41 | 0.58 | 0.56 – 0.59 |
| **Toulouse** | 0.34 | 0.31 – 0.36 | 0.40 | 0.39 – 0.44 |

The second parameter is the number $K$ of best analogs that we use to simulate the precipitation. Our choice was based on numerical experiments. We performed different SWG simulations where we varied the number of analogs $K = 5, 10, 20$. We notice an improvement on the skill scores by increasing the number of analogs as shown in table 2. Therefore, we considered

$K = 20$ analogs to ensure that we have enough analog dates for the simulations. It appears that the Euclidean distance of analogs grows rather slowly after $K = 20$. Our choice was also comforted by a theoretical study by (Platzer et al., 2021) who showed that, for complex systems, the use of a large number of analogs ($K > 30$ analogs) does not change much the prediction properties with analogs. Indeed, we kept $K = 20$ best analogs for the rest of analyses.

**Table 2.** CRPSS versus persistence and climatology for SWG simulations with 5, 10 and 20 analogs for the [30°W – 20°E ; 40° – 60°N] domain and for a lead time of 5 days.

| Location | $K = 5$ **analogs** | | $K = 10$ **analogs** | | $K = 20$ **analogs** | |
|---|---|---|---|---|---|---|
| | Persistence | climatology | Persistence | climatology | Persistence | climatology |
| **Berlin** | 0.29 | 0.20 | 0.39 | 0.31 | 0.56 | 0.50 |
| **Madrid** | 0.32 | 0.31 | 0. 40 | 0.39 | 0.57 | 0.57 |
| **Orly** | 0.34 | 0.12 | 0. 40 | 0.23 | 0.60 | 0.53 |
| **Toulouse** | 0.34 | 0.24 | 0.38 | 0.45 | 0.41 | 0.48 |

We quantify the dependence of the forecast on the time embedding for the analogs $\tau$ by calculating the analogs based on different embedding going from $\tau = 1$ to 4 days. We find that an embedding of 4 days helped to better catch the persistence and improve the skill scores for the forecast compared to 1 day, as shown in Table 3. Therefore we kept the forecast based on a 4-day embedding. This choice was based on the numerical experiments performed for the studied locations. This is also supported by the study of Yiou et al. (2013), where the analog computation with delays was argued to improve the temporal smoothness of simulations. With such an embedding, forecasts for lead times of $T = 5$ days yield at least two time increments.

**Table 3.** Correlation between observations and the median of 100 simulations for the winter (DJF) based on analogs computed with an embedding of 1 and 4 days for the geographic domain with the coordinates 30°W – 20°E ; 40° – 60°N for a lead time of 5 days.

| Location | $\tau = 1$ **day time embedding** | | $\tau = 4$ **day time embedding** | |
|---|---|---|---|---|
| | Correlation | 95% confidence interval | Correlation | 95% confidence interval |
| **Berlin** | 0.39 | 0.37 – 0. 43 | 0.50 | 0.48 – 0.56 |
| **Madrid** | 0.40 | 0.38 – 0. 42 | 0.53 | 0.51 – 0.55 |
| **Orly** | 0.42 | 0.39 – 0. 45 | 0.58 | 0.56 – 0.59 |
| **Toulouse** | 0.35 | 0.34 – 0.37 | 0.40 | 0.39 – 0.44 |

For comparison purposes, SWG simulations are obtained using analogs computed from reanalyses on the NCEP and ERA5 reanalyses. By comparing their skill scores, we found that CRPSS and correlations between observations and simulations are positive in both cases, and showing positive improvement comparing to persistence and climatology forecasts. The CRPSS and correlation for simulations with analogs of NCEP are almost identical to those with ERA5, as shown in Table 4. Therefore, we focus on SWG simulations with analogs from the NCEP reanalysis in the sequel as both NCEP and ERA5 (1950 to 2019) have the same skill, as shown in Table 4, and NCEP is easier to handle, as its horizontal resolution is much lower. The computations

were made using observations of precipitation from the ECAD (Klein Tank et al., 2002) and E-Obs (Haylock et al., 2008)
databases. We found the same results because the ECAD and E-Obs are highly correlated (by construction of E-Obs).

**Table 4.** Comparison between the values of the CRPSS of SWG computed using different reanalysis dataset NCEP and ERA5 from 1979 to 2019 for a lead time of $T = 5$ days for winter (DJF)

| Location | CRPSS DJF (ERA5) | CRPSS DJF (NCEP) |
|----------|------------------|------------------|
| Berlin   | 0.50             | 0.50             |
| Madrid   | 0.55             | 0.57             |
| Orly     | 0.53             | 0.53             |
| Toulouse | 0.41             | 0.41             |

In summary, we made the forecast of the precipitation using $K = 20$ analogs computed from Z500 over the [30°W – 20°E; 40° – 60°N] domain (red rectangle in Figure 1 b). To compute analogs, we used NCEP reanalyses and an embedding of $\tau = 4$ days. The computations were based on ECAD observations (Klein Tank et al., 2002).

## 5 Results

### 5.1 Sample forecast

As an example, we illustrate the behavior of the trajectories in Orly for the summer and winter of 2002. Figure 3 shows the observed and simulated values of precipitation for lead times of 5 and 10 days for summer (June–July–August: JJA) and winter (December–January–February: DJF), for Orly precipitation data. We observe significantly positive correlations between observed values and the median of the forecasts, for the four data sets as represented in Table 5 . The correlation is generally 245 smaller in the summer than in the winter. The correlation skill is low for some extremes values of precipitation. For a lead time of 10 days, SWG simulation still show capacity to predict precipitation especially for winter with a correlation equal to 0.23 (Orly), 0.30 (Berlin), 0.43 (Madrid), 0.31 (Toulouse).

**Table 5.** Correlation between observations and the median of 100 simulations for both seasons winter (DJF) and summer (JJA) for a lead time of 5 days

| Location | Correlation DJF | 95% confidence interval | Correlation JJA | 95% confidence interval |
|----------|-----------------|-------------------------|-----------------|-------------------------|
| Berlin   | 0.50            | 0.48 – 0.56             | 0.22            | 0.21 - 0.23             |
| Madrid   | 0.53            | 0.51 – 0.55 - 0.59      | 0.29            | 0.27 - 0.30             |
| Orly     | 0.58            | 0.56 – 0.59             | 0.23            | 0.20 - 0.24             |
| Toulouse | 0.40            | 0.39 – 0.44             | 0.18            | 0.15 - 0.19             |

We observe that the 5th and 95th quantiles of simulations include the different values of observations. This heuristically confirms the good skill of SWG to forecast precipitation from Z500 for several seasons (winter and summer) in several locations for $T = 5$ and $T = 10$ day lead times.

The difference of the forecast correlation skills between the four studied locations may be related to the variation of the local climate from one region to an other. The studied areas are in different climate types according to Köppen-Geiger's climate classification map (Peel et al., 2007). From the south western side of Europe, Madrid is in the arid zone (Peel et al., 2007), which indicates that convective rains are less significant, so that the origin of precipitation might be the result of humidity coming from the Atlantic. Conversely, Berlin is located in a cold zone characterised by warm summer and the absence of a dry season (Peel et al., 2007), so that the precipitation could be the result of both convective rains and Atlantic humidity.

In this paper, we decided (for simplicity) to use the same analogs to forecast precipitation for those four stations as discussed in section 4. A refinement of the analog regions would be necessary when focusing on Madrid vs. Berlin.

## 5.2 Forecast probability skill

The CRPSS and correlation skill scores are computed for the four studied stations Orly Berlin, Madrid and Toulouse, as shown in illustrations represented in (Figure 4) and for lead times from 5 to 20 days. We represent skill scores for January and July in order to show the skill of the SWG to predict precipitation in different conditions.

In this paper, we choose to present the results for summer and winter, to highlight the capacity of the SWG to forecast the precipitation in extreme seasons.

The CRPSS against the persistence and climatology references show positive values for lead times of up to 20 days (Figure 4). The values of CRPSS against the persistence reference (represented by squares) decrease with lead times in winter for the different studied areas, showing high values over 5 days. However for summer, we notice that the values of CRPSS against persistence increase with lead time, with high values over 20 days expect for Berlin. That indicates that for the summer until 20 days the SWG forecast is still better than the persistence forecast (the average of the CRPS of SWG is smaller than the average of the CRPS of the persistence). That could be explain by the fact that summer precipitation in Orly (51% of the time, on average) comes in clusters contrary to precipitation in Berlin. Indeed, we computed the seasonal frequency of precipitation (defined as the number of days when precipitation exceeds 0.5 mm/day). We found that for Berlin, precipitation exceeding 0.5 mm/day is more frequent than in the other stations (close to 50% of the time for both seasons).

This means that a persistence forecast for Orly is likely to be skillful, even for longer lead times, especially in the summer. Therefore, the trends in CRPSS values for different lead times are probably due to the intrinsic time persistence of local precipitation.

The CRPSS against climatology reference (triangles) show lower values compared to the CRPSS against persistence reference, although they are positive for all lead times and for both seasons. However, we notice that for short lead time the SWG is better than the climatology.

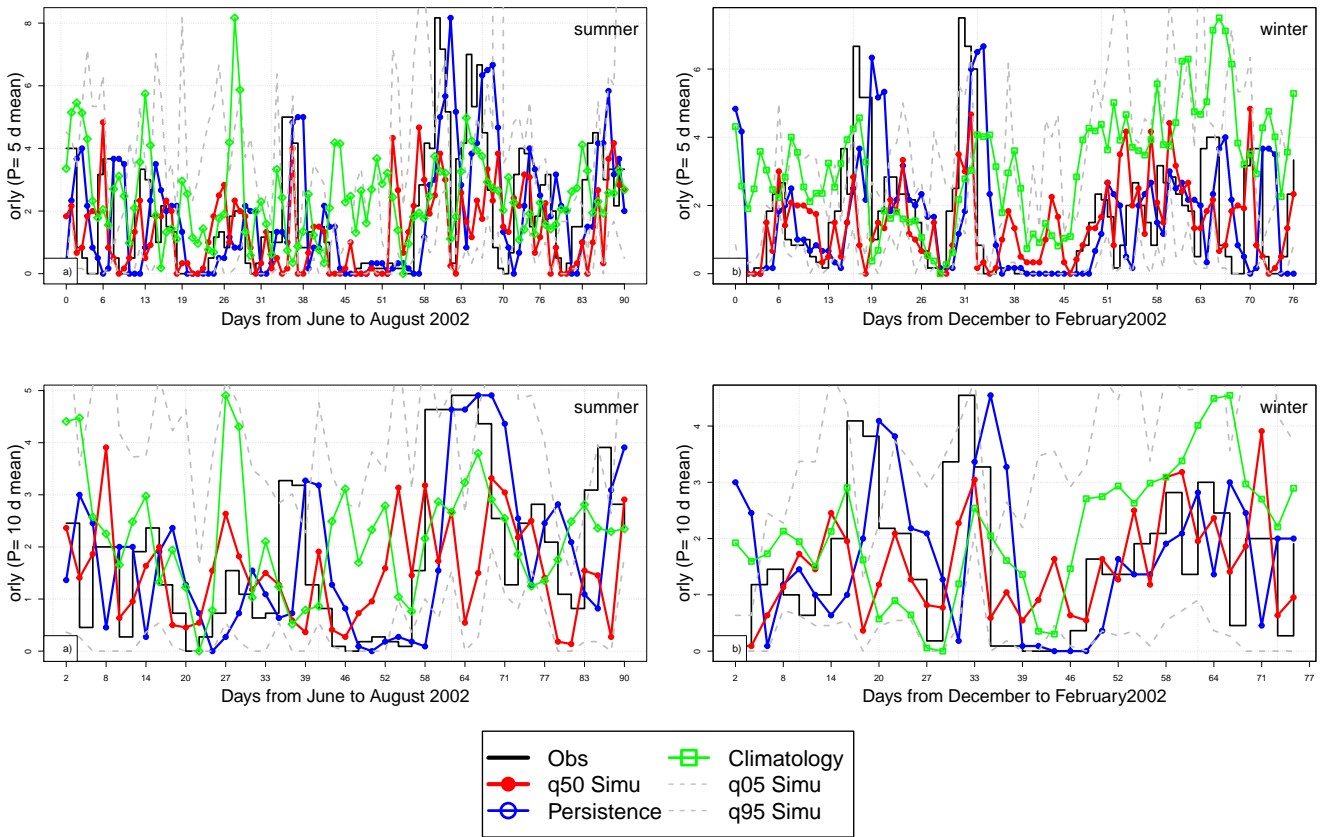

**Figure 3.** Time series of analog ensemble forecasts for 2002, for lead times of 5 days (top) and 10 days (bottom) for summer (June to August) a) and c) and winter (December to February) b) and d) for Orly. The median of 100 simulations is represented by red line. Black line represent observations values. Gray lines represent the 5th and 95th quantiles. Blue lines represent persistence forecasts and green lines represent the climatology forecasts. The y-axis represent the average of precipitation over $T = 5, 10$ days.

The correlation skill is positive for both seasons but higher in winter (January) than in summer (July). For a lead time of 5 days, the correlation is equal to 0.59 for Madrid, 0.50 for Berlin and to 0.40 for Toulouse. For a lead time of 10 days, it is equal to 0.42 for Madrid, 0.30 for Berlin and to 0.41 for Toulouse.

The SWG was tested by Yiou and Déandréis (2019) to forecast temperature in western Europe. Comparing the performance of the SWG to forecast those different meteorologic variables, we notice that the model shows good performance to forecast the temperature in the winter, also the best performance of the model is at a lead time of 5 days. We find that the skill scores (CRPSS and correlation) decrease with lead times. The forecast skill of the SWG shows variability from one location to another. However, the model was able to forecast temperature until 40 days in Berlin, Orly and Toulouse with positive skill scores.

From a visual inspection of the CRPSS and correlations, we chose to focus on lead times of $T = 5$ days, for which the correlation exceeds 0.5 in the winter. It is rather low in the summer, due to convective events leading to a high precipitation

variability (from no rain to very high values). Correlation scores become barely significant for lead times of 20 days, so that, like temperature, the SWG should not be used beyond that horizon.

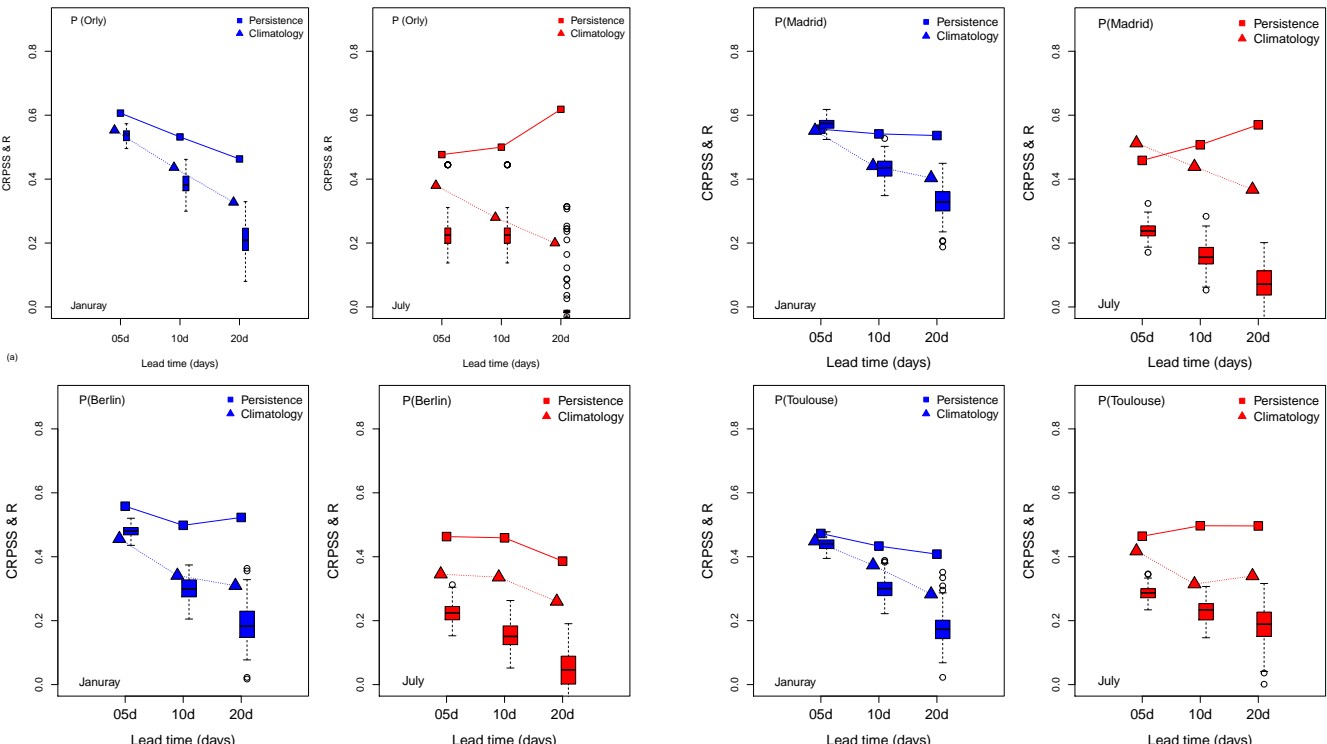

**Figure 4.** Skill scores for the precipitation of Orly, Madrid, Berlin and Toulouse for lead times of 5, 10, 20 days for January (blue) and July (red) for analogs computed from reanalyses of (a) NCEP and (b) ERA5. Squares indicate CRPSS where the Persistence is the baseline, triangles indicates CRPSS where the climatology is the reference, and boxplots indicates the probability distribution of correlation between observation and the median of 100 simulations for all days. The boxplot upper whisker is: $\min\{1.5(q_{75} - q_{50}) + q_{50}, \max(CRPS)\}$. The boxplot lower whisker is: $\max\{q_{50} - 1.5(q_{75} - q_{50}), \min(CRPS)\}$.

### 5.3   Relation between weather regimes and CRPS

We investigate the role of North Atlantic weather patterns defined in Section 3.4 (Figure 2) on the forecast skill of the SWG precipitation simulations.

We start by comparing the frequencies of the weather regimes from the observations and the most frequent weather regime found in SWG simulations for a given lead time $T = 5$ days. We find that the percentages are very similar (Figure 5). This means that the weather regimes of the simulated trajectories do not yield major biases for the summer or winter seasons.

Then we look at the relation between weather regime and the CRPS, by using the most frequent weather regime and the two classes of quantiles of the CRPS that related to good quality of forecast (attributed to low values of CRPS $\leq q_{25}$) and poor

quality of forecast (attributed to high values of CRPS $\geq q_{75}$). This relation is represented in Figure 6 for Orly and for the rest of the studied stations in Figure A1. We find a small influence of specific weather regimes on the CRPS distribution for summer.

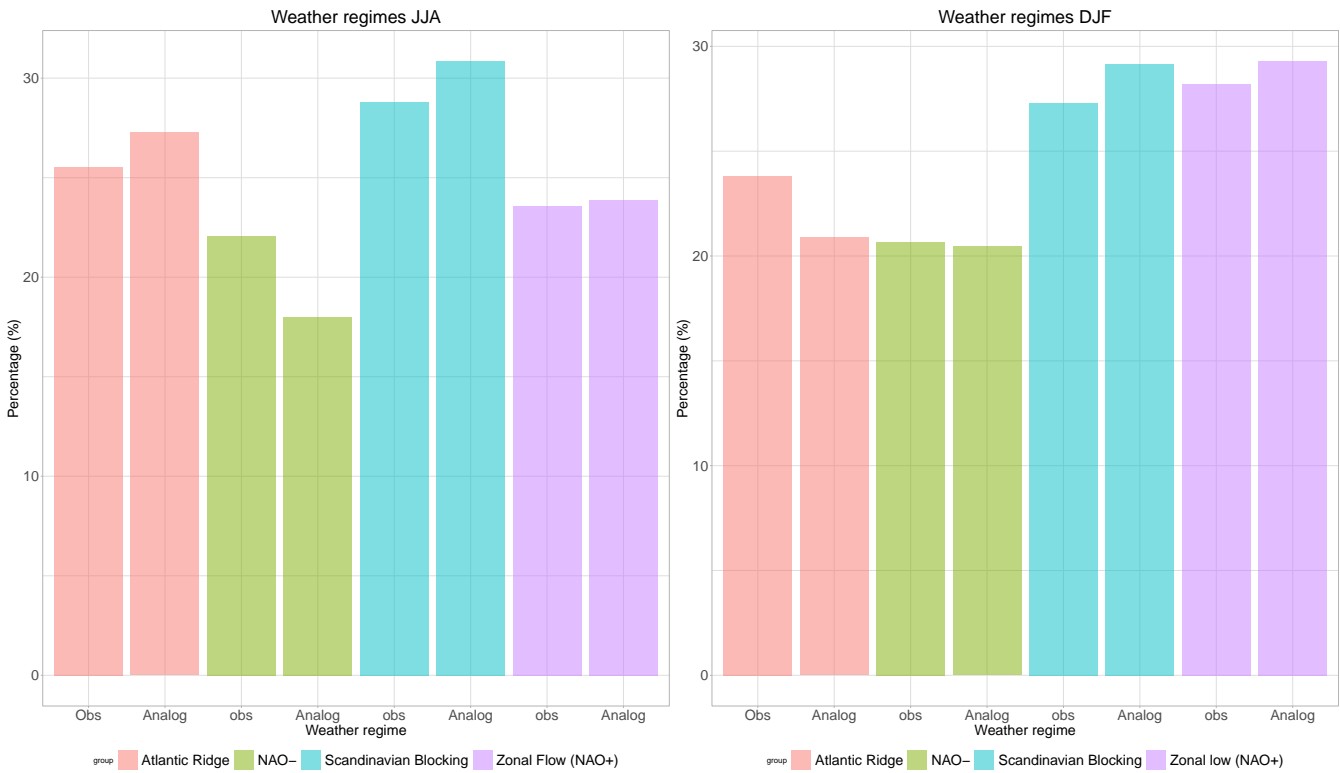

**Figure 5.** Percentage of each weather regime for observations dates (Obs) and the most frequent weather regime from SWG simulations between $t_0$ and $t_0 + T = 5$ days (Analog) over the period from 1948 to 2019 for summer (JJA) and winter (DJF). The percentage of weather regime are the same in Obs and Analog.

The weather regime signal for "good" forecasts depends on the season and the considered station. When the forecast has a low CRPS value (for Orly), we find that the Scandinavian Blocking regime slightly dominates (green bar in Figure 6a, b). This is also the case for Berlin (in winter) and Toulouse Figure A1 (b, j). The low CRPS values in Madrid are obtained for the

305 Atlantic Ridge regime Figure A1 f.

The weather regime signal for "poor" forecasts also yields a dependence on the season and station. Higher CRPS values are obtained with the Atlantic Ridge regime in the summer for Orly (red line in Figure 6 c) and Berlin in winter and summer. The Atlantic ridge regime favors high CRPS values (i.e. poor forecasts) for Madrid in winter Figure A1 h. The Atlantic ridge regime favors high CRPS values for Toulouse in summer. The different impacts of the weather regimes on the studied areas

is related to the position of the high and low pressure regions of each weather regime and their position regarding the studied areas.

    This relation between predictability (or the CRPS distribution) and weather regimes, albeit weak, is consistent with previous work of Faranda et al. (2017). Similar relation were found between weather regimes over Europe and the Temperature in a recent study by Ardilouze et al. (2021) . We find that the sensitivity of the forecast to weather regime is larger for low values of

CRPS and in the winter. The sensitivity of forecast skill to weather regimes is rather small on average, even for low lead times.

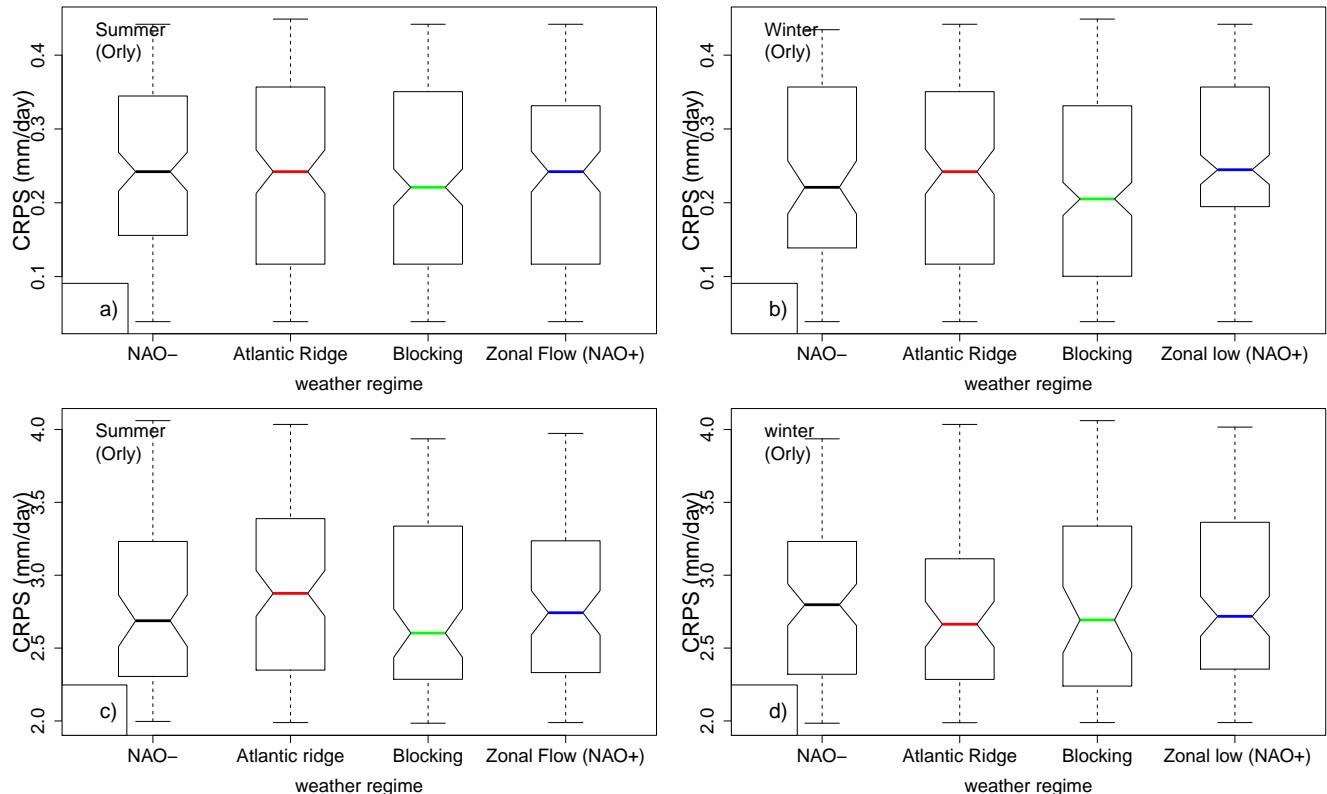

**Figure 6.** Relation between CRPS and weather regimes for Orly, for SWG forecasts with lead time $T = 5$ days. Upper panels (a and b): CRPS value distribution conditioned on four weather regimes, when CRPS is lower than $q_{25}$. lower panels (c and d): CRPS is higher than $q_{75}$. The boxplots indicate the median ($q_{50}$) of the distribution (thick bar). 25th ($q_{25}$) and 75th ($q_{75}$) quartiles (lower and upper segments). The boxplot upper whisker is: $\min\{1.5(q_{75} - q_{50}) + q_{50}, \max(CRPS)\}$. The boxplot lower whisker is: $\max\{q_{50} - 1.5(q_{75} - q_{50}), \min(CRPS)\}$.

### 5.4   Comparison with ECMWF forecast

We first compared the CRPSS of SWG forecasts for winter and summer with the CRPSS of ECMWF forecasts.

    The CRPSS of ECMWF forecast is computed for different lead times going from 1 day to 10 day for precipitation (Haiden et al., 2018) over the region 12.5°W – 42.5°E ; 35.0° – 75.0°N (ECMWF, 2020). It uses the climatology as a reference

(Haiden et al., 2018). The values of CRPSS for Europe for 2020 decrease with lead times (Haiden et al., 2018). The CRPSS of ECMWF is about 0.16 in summer (JJA) and 0.25 in winter (DJF) for a lead time of $T = 5$ days (ECMWF, 2020). The CRPSS of SWG simulations for a lead time of $T = 5$ days is shown in Table 4. The values suggest that the predictive skill of SWG is qualitatively promising for short lead times, compared with ECMWF forecasts. However, we have to mention that the values of CRPSS for ECMWF are computed over all Europe for both seasons (Haiden et al., 2018) while with the SWG we are doing

forecast for local stations.

We made a quantitative comparison between the two forecasts for the different lead times. We computed the CRPS for the ECMWF forcast. Then, we used the Kolmogorov-Smirnov (KS) test (von Storch and Zwiers, 2001, Chap.1) to compare the probability distributions of the CRPS of SWG and ECMWF forecasts. The null hypothesis supposes that the CRPS of ECMWF and SWG forecasts have the same distribution. The null hypothesis of KS test was rejected, which means that the two times

330 series do not have the same distribution with a p.values $= 0.11$. A similar result was found by Ardilouze et al. (2021), where they compared the efficiency between ECMWF and CNRM forecasts.

We found that 80%, 39% 50% and 40 % of the CRPS of SWG forecast are equal to zero for respectively Orly, Berlin, Madrid and Toulouse, for a lead time of $T = 5$ days (Figure 7), which shows the capacity of the SWG to well simulate rain events. One notable difference between SWG and ECMWF forecasts is that although the proportion of CRPS values close to zero

is higher in ECMWF, the CRPS for the worst forecasts are much higher than those of SWG. Indeed, we notice that the time average of CRPS of ECMWF (blue vertical lines) and SWG (red vertical lines) for $T = 5$ days are close, with higher values for ECMWF 7. However, the median of CRPS of ECMWF are smaller compared to the SWG (dashed vertical lines) 7. Finally, we computed the CRPSS for ECMWF forecasts taking as a reference the CRPS of SWG (Table 6). We hence compute the CRPSS of ECMWF forecast by normalizing the CRPS by the CRPS of SWG forecast in Eq. (C1).

**Table 6.** CRPSS of ECMWF forecasts using as a reference the CRPS of SWG, for lead times of $T = 5$, 10 and 20 days. It shows that the SWG has a positive improvement comparing to the ECMWF forecast as the CRPSS are above zero, expect for Toulouse.

| Location | Orly | Berlin | Madrid | Toulouse |
|---|---|---|---|---|
| **CRPSS $T = 5$ days** | -0.09 | -0.02 | -0.2 | 0.25 |
| **CRPSS $T = 10$ days** | -0.17 | -0.54 | -0.33 | 0.23 |
| **CRPSS $T = 20$ days** | -0.50 | -0.36 | -0.1 | -0.08 |

This evaluates the added value of the ECMWF forecast over the SWG forecast. We find that the ECMWF forecast has no improvement over the SWG forecast for the different lead times because the CRPSS value are negative. At $T = 5$ days, we notice that the improvement is negligible for Orly and Berlin, while it is much better for Madrid. However for Toulouse, the ECMWF forecast still have better skills for lead times of $T = 5, 10$ days. For a lead time of $T = 10$ days, the improvement of SWG forecast over the ECMWF is important especially for Berlin and Madrid. There is a major improvement for a lead

time of $T = 20$ days for Orly and Berlin. This confirm the relatively good skill of the SWG to forecast precipitation, compared to ECMWF. And that could be explained by the difference on the average of the CRPS of the two forecasts. Indeed, as we

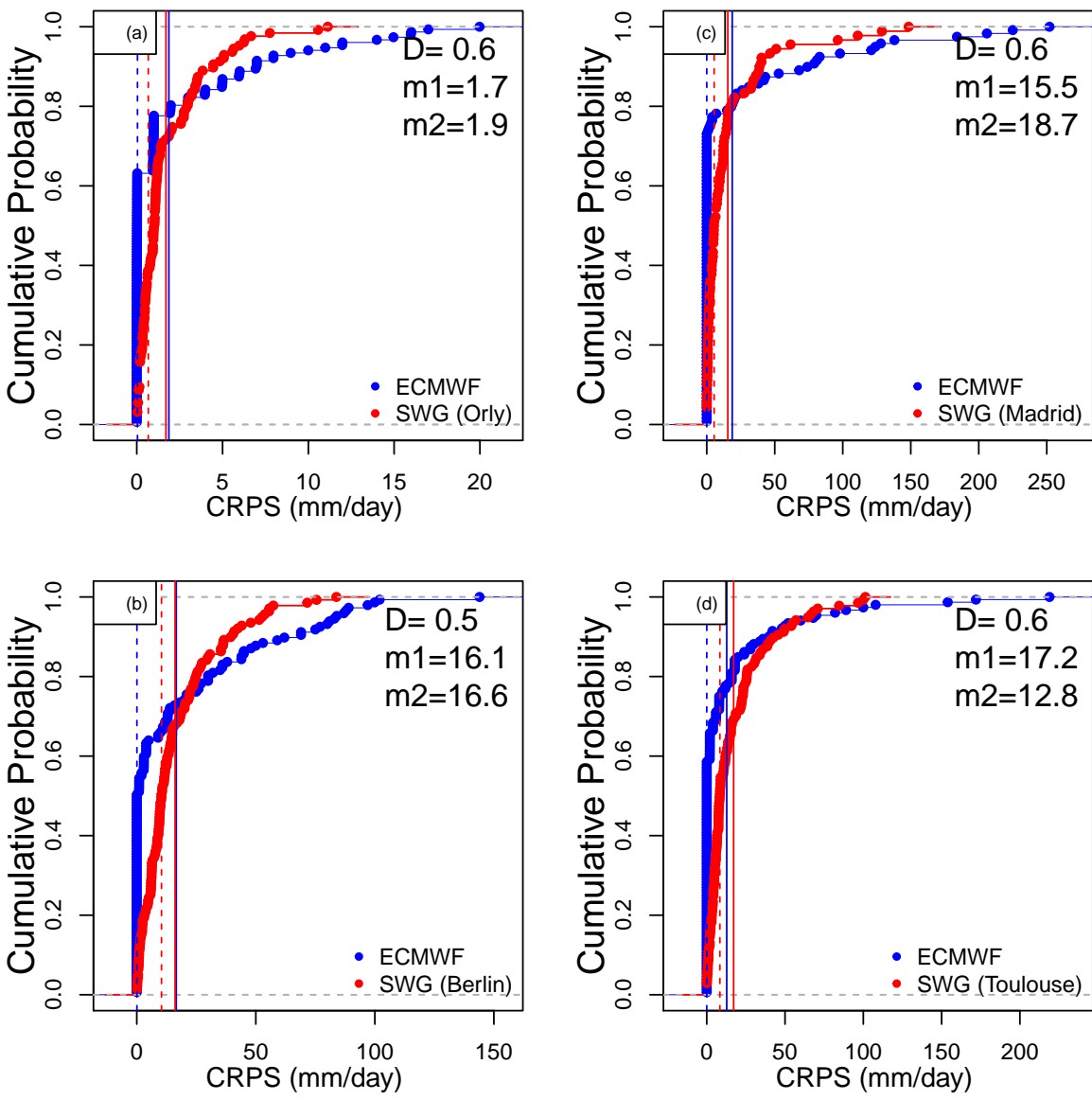

**Figure 7.** Empirical cumulative distribution function of the CRPS of ECMWF (blue) and SWG (red) forecasts for 5 days for Orly (a), Berlin (b), Madrid (c) and Toulouse (d). D is the maximum distance between both ECDFs (value of Kolmogorov-Smirnov test). $m1$ is the value of the time average of CRPS of SWG and $m2$ is the value of the time average of CRPS of ECMWF. The vertical dashed lines represent the median of CRPS of ECMWF (blue) and SWG (red).

mentioned before, the ECMWF forecast has good skill for small values of precipitations, we further explained that on Annexes (Figure C1 and Table C1).

## 6 Conclusions

In this work, we have shown the performance of a stochastic weather generator (SWG) to simulate precipitation over different locations in western Europe and for various times scales from 5 to 20 days. The input of our model was analogs of geopotential heights at 500 hPa (Z500). The choice of such input was made in order to evaluate the impact of large scale circulation on local weather variables. SWG showed a good skill to predict precipitation for a lead time of 5 and 10 days from analogues of Z500.

This study complements the work of Yiou and Déandréis (2019), for precipitation. We explored the sensitivity of the SWG model on analogs computed from different geographical areas and from different reanalyses (ERA5 and NCEP). We found that the NCEP and ERA5 reanalyses provide good performances for simulations, due to its longer length ($\approx 70$ years in NCEP and ERA5). Therefore the length of the analog database does make a difference, as already suggested by Jézéquel et al. (2018a).

We evaluated the relation between the quality of the forecast and weather regimes over Europe, we found that low and high predictability was related to specific weather regimes, this dependence is more significant in winter than in summer, especially for the good predictability, it is found to be mainly related to Blocking.

A comparison with the ECMWF forecast system over Western Europe confirmed the good performance of the SWG quantitatively and qualitatively, for lead times $T \leq 10$ days. Of course, the SWG model cannot replace a numerical weather prediction, as the SWG parameters (e.g. region of analogs) need to be tuned to local variables, and rely on the existence of a fairly large database to compute analogs. Here we used the same domain of circulation analogs for stations from Madrid to Berlin. Obviously, this region should be optimized for each individual station. Therefore, the main utility of the SWG forecast system is to make local ensemble simulations, where its performances can challenge a numerical weather prediction, if the parameters are well tuned.

This paper hence confirms the proof of concept to generate ensembles of (local) precipitation forecasts from analogs of circulation. Its performance relies on the relation between precipitation and the synoptic atmospheric circulation, which is verified for western Europe. Transposing this SWG to other regions of the globe requires observations covering several decades. Numerical weather models obviously do not yield this constraint.

*Code availability.* The code and data files are available at http://doi.org/10.5281/zenodo.4524562

*Author contributions.* MK performed the analyses. PY co-designed the analyses. CD and ST participated to the manuscript preparation.

*Competing interests.* The authors declare no competing interest.

*Acknowledgements.* This work is part of the EU International Training Network (ITN) Climate Advanced Forecasting of subseasonal Extremes (CAFE). The project receives funding from the European Union's Horizon 2020 research and innovation programme under the Marie Skłodowska-Curie Grant Agreement No 813844. We thank L. Magnusson and F. Pappenberger for helpful discussions on the ECMWF data.

## Appendix A: CRPS and weather regimes

To avoid a tedious redundancy we deferred the figures of evaluation of the forecast quality by weather regimes to this appendix
section.

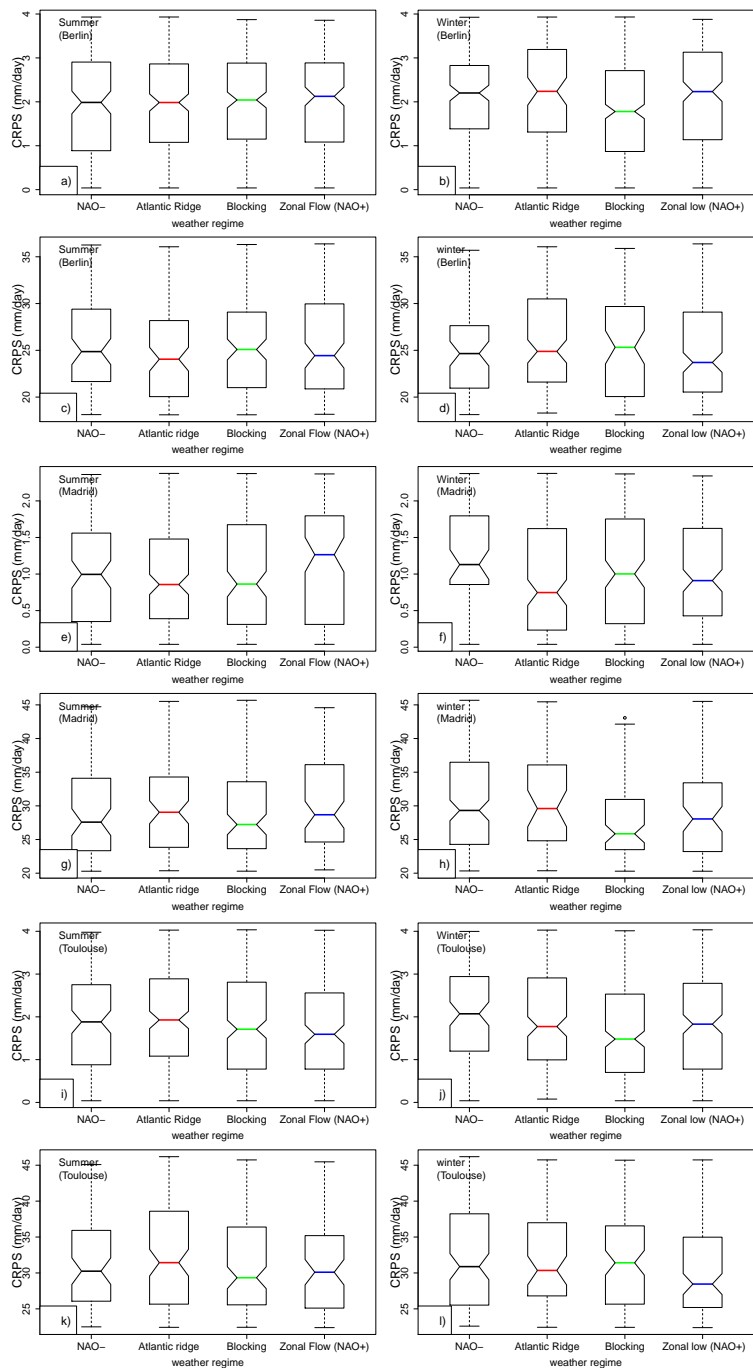

**Figure A1.** Relation between CRPS and weather regimes for Berlin (a–d)), Madrid (e–h) and Toulouse (i–l), for SWG forecasts with lead time $T = 5$ days. The panels (a, b, e, f, i and j) correspond to CRPS value distribution conditioned on four weather regimes, when CRPS is lower than $q_{25}$. The panels (c, d , g, h, k and l) correspond to higher CRPS value $CRPS \geq q_{75}$. The boxplots indicate the median ($q_{50}$) of the distribution (thick bar).

## Appendix B:  Relation between Z500 and precipitation

In order to justify, the use of the Z500 as driver of precipitation. We computed the rank spatial correlation between the daily average of Z500 over the Euro Atlantic region and the precipitation in each studied station (Madrid, Berlin,Toulouse and Orly). We did the analysis for different seasons (DJF - JJA). We find a maximum correlation amplitude of -0.5 for Madrid and Orly, and a correlation of -0.4 and -0.3 respectively for Toulouse and Berlin. The correlation is significant as we have a p.value $< 0.05$ for the different grid points. This indicates the relation between Z500 patterns and precipitation especially in western Europe and that a decrease in Z500 is linked with precipitation.

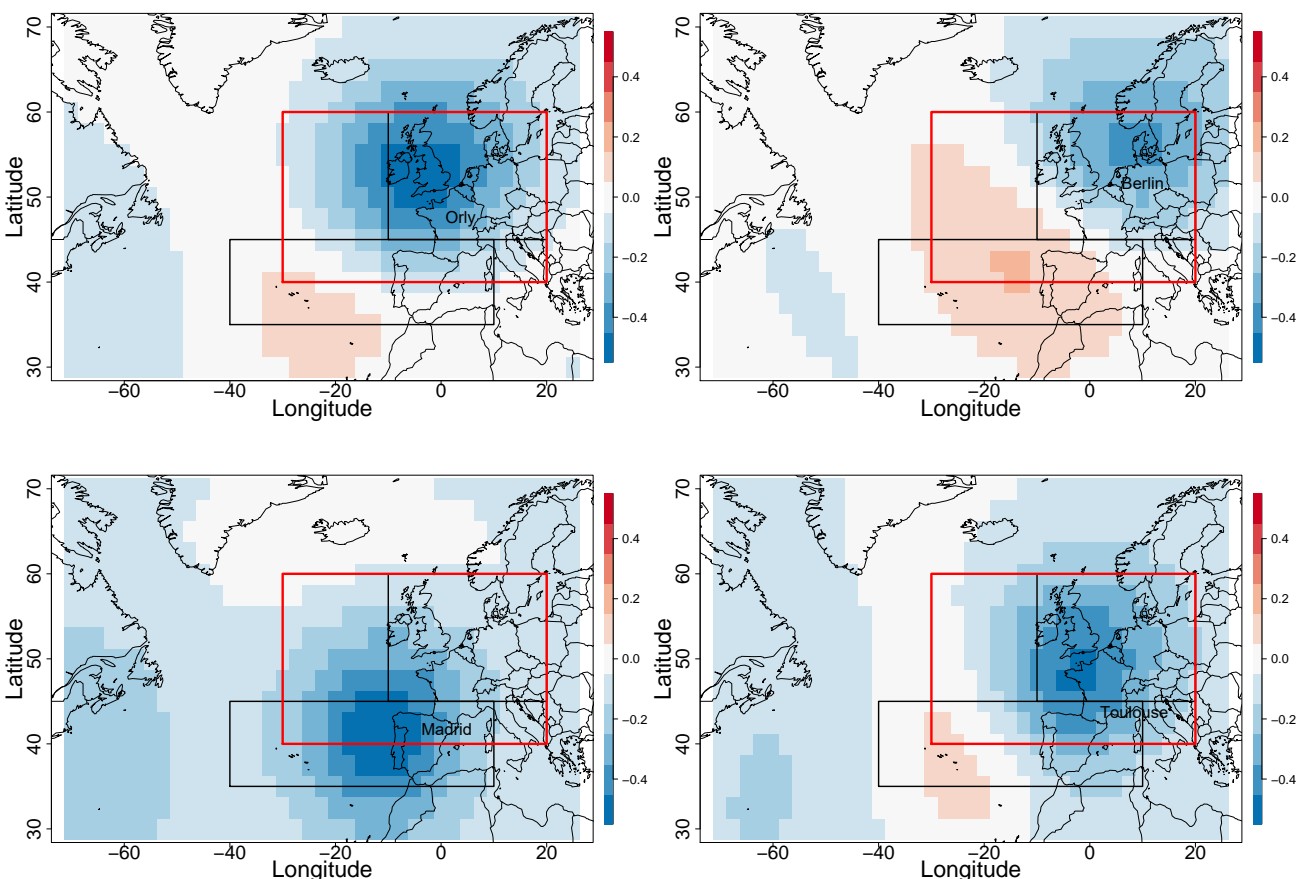

**Figure B1.** Maps of correlation between Z500 and precipitation in Berlin, Madrid, Orly and Toulouse for the period from 1948 to 2019 over the Euro-Atlantic region. The rectangles represent the domains of computation of analogs. We highlight the optimized area [30°W-20°E; 40°-60°N] by the red rectangle.

## Appendix C:  CRPSS of ECMWF versus SWG

We explain further the comparison that we made between ECMWF forecast and SWG forecast. As mentioned we found that
the SWG has improvement comparing to ECMWF forecast. This is related to the difference on the time average of the CRPS of the two forecasts. We computed the CRPSS as follow:

$$CRPSS = 1 - \frac{\overline{CRPS_{ECMWF}}}{\overline{CRPS_{SWG}}} \tag{C1}$$

with $\overline{CRPS_{ECMWF}}$ is the time average of the $CRPS$ of the ECMWF forecast and $\overline{CRPS_{SWG}}$ is the time average of the $CRPS$ of the SWG.

**Table C1.** CRPSS, average and median of CRPS of ECMWF and SWG forecasts for lead times of $T = 5$, 10 and 20 days. It shows that the CRPS of SWG forecast has a smaller average than the CRPS of ECMWF forcast, which explains the values of CRPSS for the different studied areas and the positive improvement of the SWG compared to ECMWF.

| Location | Orly | Berlin | Madrid | Toulouse |
|---|---|---|---|---|
| $\overline{CRPS_{ECMWF}}$ / Median | 1.87 / 0.04 | 16.56 / 0.05 | 18.73 / 0.003 | 12.76 / 0.01 |
| $\overline{CRPS_{SWG}}$ / Median | 1.70 / 0.67 | 16.10 / 10.37 | 15.49 / 5.45 | 17.16 / 8.39 |
| **CRPSS $T = 5$ days** | **-0.09** | **-0.02** | **-0.2** | **0.25** |
| $\overline{CRPS_{ECMWF}}$ | 1.70 / 0.05 | 18.1 / 0.06 | 20.03 / 0.1 | 14.87 / 0.09 |
| $\overline{CRPS_{SWG}}$ | 1.44 / 0.78 | 11.67 / 5.45 | 15.04 / 6.13 | 19.45 / 7.89 |
| **CRPSS $T = 10$ days** | **-0.17** | **-0.54** | **-0.33** | **0.23** |
| $\overline{CRPS_{ECMWF}}$ | 1.67 / 0.1 | 13.54 / 0.09 | 17.89 / 0.1 | 17.8 / 0.08 |
| $\overline{CRPS_{SWG}}$ | 1.11 / 0.9 | 9.91 / 6.3 | 16.23 / 5.89 | 16.41 / 8.34 |
| **CRPSS $T = 20$ days** | **-0.50** | **-0.36** | **-0.1** | **-0.08** |

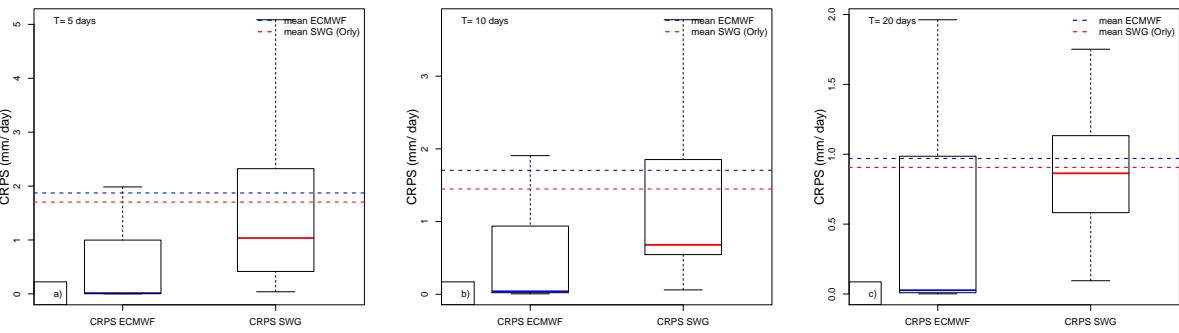

**Figure C1.** Boxplots of CRPS of ECMWF and CRPS of SWG for Orly, with lead time $T = 5, 10, 20$ days. The boxplots indicate the median ($q_{50}$) of the distribution (thick bar blue for ECMWF and red for SWG). 25th ($q_{25}$) and 75th ($q_{75}$) quartiles (lower and upper segments). The upper whisker is $\min\{\max(X), q_{50} + 1.5(q_{75} - q_{25})\}$. We indicates also the average of CRPS of ECMWF and SWG forecasts with horizontal dash lines. We notice clearly that the distribution is asymmetric as the median and the average are unequal. And that the average of CRPS for SWG forecast is lower than the average of CRPS for ECMWF forecast. We do not show the outliers that are above the upper whiskers.

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
