# Peer review of "Assessment of stochastic weather forecast of precipitation near European cities, based on analogs of circulation"

_Geoscientific Model Development, 2021_

## Referee Comment (RC1)

**Review of « Assessment of stochastic wether forecast of precipitation near European cities, based on analogs of circulation » by Krouma et al.**

**General comments :**

The article assesses the skill of a stochastic weather generator to forecast precipitation in 4 cities of Western Europe. The SWG is based on random sampling of analogs of geopotential height. It was developed in another article by Yiou and Déandréis 2019, where it was applied to temperature. This study complements the latter for precipitation. As a refinement, a time embedding of 4 days is considered in the distance for the search of analogs (however the considred distance is not a mathematical distance anymore - see below). Skill scores are evaluated for lead times of 5 to 20 days. Results show positive skills up to 10 days. A comparison to ECMWF forecasts is provided but I have some concerns about this part (see below).

The study is interesting, clear and well written. Precipitation forecasting is an important subject of research and I support the idea basing SWG on analogs. However I have several main concerns :

- The results are mainly shown for NCEP which used to cover a longer period than ERA5. However ERA5 is now available since 1950. Given its much better resolution, I recommend considering ERA5 for all the results.

- I'm surprised that all the applied tests  (Table 1 and p 14)  have pvalues equal to $2.2^{-16}$ (I guess you mean $2.2 \cdot 10^{-16}$?). Isn't that strange ? More importantly, I doubt that Kolmogorov Smirnov gives such low pvalues given the differences in the CDF of Figure 6.

-  I'm concerned about the comparison with ECMWF forecasts since ECMWF are gridded data, whereas SWG is based on point data (ECAD). Have you considered comparing ECMWF forecasts with SWG based on E-OBS, since both have an horizontal resolution of 0.25°x0.25° ?

- The evaluation of CRPSS conditional on weather regimes is interesting but I wonder whereas considering the weather regime of the last day of the sequence (t0+T) is representative for the weather regime of the whole sequence.

More minor, some references are missing (see below). There are issues in the units of CRPS. There are several equation issues.

**Detailed comments :**

- l 65 : Reference to Klein Tank is missing
- l 67 : please specify that ECAD provides point (station) data
- l 72-81 : actually ERA5 is now available since 1950.
- l 84 : Reference to Herschbach is missing. By the way, is it the right reference ?
- l 87 : so ECMWF forecasts have the same resolution as EOBS.
- l 104 eq (1) : This is a good idea to account for several days in the distance, however D in (1) is not anymore a mathematical distance. Of course this is not mandatory for analog search, however why not using [sum_x sum_i {Z500(x,t+i)-Z500(x,t'+i)²} ], which is a mathematical distance? By the way, could you pelase provide a comparison of the results with the Euclidean distance (based on 1 day) vs. the D distance (based on 4 days)? And why 4 days ?
- l 111-118 : explanations are quite confusing. I had to read Yiou and Déandréis to understand. Please consider rewritting the method.
- Figure 1 : a)  please consider placing the red rectangle somewhere else within the 30 days for clarity since its date is not necessarily the same as the target day. b) the largest window doesn't match the coordinates given l 80. I would be happy to see some results on the other windows of analogy. Otherwise I think it's not worth showing them. Also there are several syntax issues in the caption
- l 122 : please specify that persistence is computed over year k (unlike the climatology which is computed over all years)
- l 124 : « control forecast » I dont't understand

- l 137 : I guess that averaging the 100 trajectories smooths out the predictions. So at the end, is there a real gain (in terms of CRPSS) compared to considering only one analog ? (maybe that's already studied in another article, I haven't checked)

- l 150 : P(x) should be P(x,t) for day t. Please also rephrase the sentence

- l 153 eq (2) : the equation is confusing. Should be CRPS(P,t) and t should be in the right side as well. The inferiori limit is 0 for precipitation.

- l 159 : seasonality → climatology

- l 162 eq (3) : Equation issues. there should a sum over the days (or mean) in the numerator and denominator

- Table 1 : is it Pearson correlation ? I'm surprised that all pvalues equal $2.2^{-16}$ (I guess you mean $2.2 \times 10^{-16}$?).

- l 198-201 : this paragraph should go after l 204. Please refer to Fig 3.

- l 212 : syntax issue

- Table 2 : I guess this is for NCEP ?

- l 228 : remove the brackets

- l 230 : so you obtain 100 classifications. How do you deal with that ?

- l 241 : is the weather regime at time t0+T representative of the sequence from t0 to t0+T. Why don't you consider the most frequent WR within t0 to t0+T ?

- Fig 3 : please consider plotting both reanalyses on the same plot for ease of comparison. Actually for persistence ERA5 seems to give larger CRPSS. Caption : persistence in lowercase letter. Please use either « reeference » or « baseline » along the article. For the boxplots, why don't you consider correlation with the mean instead of the median (as the predictions of SWG) ? (but it should not change much)

- l 243 : don't you mean below the 25th quantile ? Where is this used ?

- Figure 4 caption : blank space after (BLO)

- l 251 and followings : I see there are differences depending on the WR but it seems to depend on the city.  Can we have explanations why, e.g. CRPS for BLO is beteer in Orly ?

- Figure 5 : Units of CRPS are not mm. What is the lead time here ? Given l 243-244, I would have expected here to see boxplots for the two classes of predictability.

- l 257 : what is the reference for CRPSS ? (I guess climatology)

- l 261 : do I understand correctly that in ECMWF forecasts, CRPSS is given for the whole of Europe whereas CRPS are avalaible at every grid point ? As said above, I find difficult comparing the skills of ECMWF vs SWG given that the horizontal resulution is different (0.25°x0.25° vs point data). Comparison of ECMWF with EOBS at the same resolution may be easier .

- Table 3 : please specify the reference. You may want to add here the CRPSS of Europe with ECMWF.

- l 264 : CPRSS are actually hard to compare since they are not based on the same data (different resolution)

- l 266 « We found... » I don't understand the sentence (syntax issues). Anyway according to the CDF of CRPS in Fig 6, ECMWF seems significantly better (a much larger proportion of low values)

- l 270 again $2.2^{-16}$ ? Anyway, I think something's wrong here because the CDFs in Fig 6 do seem different. A difference of 0.2 between CDFs is large actually.

- l 276 and Fig 7 : I think something's wrong because ECMWF shows a much larger proportion of small CRPS for Toulouse and Madrid (see Fig 6). The difference in CRPSS for Orly between 5 and 10 days is very surprising.

- l 300 : designed

- Some references are missing. There is no year for Cassou.

---

## Referee Comment (RC2)

**Lines 79-80 and figure 1. Why exactly that region? Is there a process-knowledge approach behind this choice, a literature review or a regression/correlation between Z500 and rain over each station was applied? Whatever, the choice, it should be justified by means of references or, in the last case, with a graph/map that certifies the link between Z500 and precipitation.**

**Line 98. Why did the author choose exactly 20 analogues? Would not be better to base the choice on a maximum Euclidean distance?**

**Line 100. The 4-day time embedding is not clear to me. Why is it necessary? Why does it preserve the temporal derivative of the atmospheric field? Please explain.**

**Lines 113-118. Despite the mechanism is quite clear to me, the sentences "In order to go [...] precipitation between $t_0$ and $t_0 + T$" are not well formulated. Since this is a crucial part for the understanding of the method, I would rephrase and expand this part.**

Line 120. "of the properties" is redundant.

Line 123. More than the average value, the persistence consists in the *anomaly* between $t_0 - T$ and $t_0$. Also, the climatological forecast takes.

**Lines 128-134. Again, it is not clear upon which basis those domains (in Fig. 1b) have been chosen and the final domain selected among the four attempts. Also, the entire paragraph needs a language revision.**

Line 150. A verb is missing (meet?).

Figure 2. A visual legend is needed. Also, what's in the y-axis? Are the 5[th] and 95[th] percentiles calculated over the 1948-2019 time series?

Line 184: six??

Lines 201-205. It seems to me that part of the methodology is described here, where a description of results is expected.

Line 206 and following. Very little description is given for figure 3. First of all, I think it is useful to illustrate all the stations in the main text, instead of showing only Orly while relegating the others to the Appendix. This is one of the main results of the study and deserves a better stage (instead, I would recommend to place Fig. 4 the Appendix, since the WRs are not a result of this study). Besides, fig. 3 and fig. A1 show a very interesting characteristic that should be discussed: most of the times, in fact, the *summer* SWG forecast vs. persistence improves with lead times, which is somewhat unexpected. Any thoughts about it?

Line 217. De Bilt?

Line 225. They help describe.

Line 232. Missing year for the Cassou reference.

Lines 246-255. Fig. 5 (B1) should be described with more care. What do "Good forecasts (low quantiles of CRPS)" and "The low quality forecasts (high quantiles of CRPS)" mean? The caption for figure 5 is totally unclear and does not describe the plot. On the top of the blocking bar (panel b) a group of dots appear.

Lines 257-276. This paragraph is hard to read, there are many inconsistencies between subjects and predicates and other grammar errors. Also, the first paragraph is not clear: what is it meant to demonstrate? Maybe the lower skill of the ECMWF forecast? The latter is calculated over the entire European domain, how can it be compared with a forecast over single stations? A table with both ECMWF and SWG CRPSS would be more informative than a few words, if the authors find the way to make a fair comparison between the two.

Line 279. *The input of our model was analogs of geopotential heights at 500 hPa (Z500).* This sentence should be rephrased.

**Line 283. I cannot accept this conclusion. The only way to test it is to compute the analogs with the 70-year ERA5 dataset, now available since 1950. This is a very important test that should be included in this study, because it clarifies the role of the different reanalyses as well as the role of database length.**

---

## Author Comment (AC1)

**Review of « Assessment of stochastic weather forecast of precipitation near European cities, based on analogs of circulation » by Krouma et al.**

⇒ **We thank the reviewer for the positive and constructive comments.**

General comments :

The article assesses the skill of a stochastic weather generator to forecast precipitation in 4 cities of Western Europe. The SWG is based on random sampling of analogs of geopotential height. It was developed in another article by Yiou and Déandréis 2019, where it was applied to temperature. This study complements the latter for precipitation. As a refinement, a time embedding of 4 days is considered in the distance for the search of analogs (however the considered distance is not a mathematical distance anymore - see below). Skill scores are evaluated for lead times of 5 to 20 days. Results show positive skills up to 10 days. A comparison to ECMWF forecasts is provided but I have some concerns about this part (see below). The study is interesting, clear and well written. Precipitation forecasting is an important subject of research and I support the idea basing SWG on analogs. However I have several main concerns :

- The results are mainly shown for NCEP which used to cover a longer period than ERA5. However ERA5 is now available since 1950. Given its much better resolution, I recommend considering ERA5 for all the results.

⇒ **We will extend the search of analogs to 1950 using the ERA5 database. Our idea was just to verify the robustness of our methodology to compute analog from different sources. We mention that the new database of ERA5 from 1950 to 1978 is still preliminary.**

- I'm surprised that all the applied tests (Table 1 and p 14) have pvalues equal to $2.2^{-16}$ (I guess you mean $2.2 \times 10^{-16}$?). Isn't that strange ? More importantly, I doubt that Kolmogorov Smirnov gives such low pvalues given the differences in the CDF of Figure 6.

⇒ **The pvalues of table 1 are obtained from the correlation test where the null hypothesis is that the correlation is equal to 0. Given the correlations observed, it is not unreasonable that the p-values of the tests are this low. However, we will instead use a confidence interval in the revision, since they provide the uncertainty around the estimate which is more appropriate in this case. We will use a confidence interval. For the Kolmogorov Smirnov (K-S) test, the p-values were given by the K-S test. In fact the K-S test with low p-values meant the rejection of the null hypothesis that the two distributions are equal. Hence, the Figure 6 and the conclusion of the KS test are consistent. However we can add the D value between CDF as a better indicator and the Student test between median of distributions.**

- I'm concerned about the comparison with ECMWF forecasts since ECMWF are gridded data, whereas SWG is based on point data (ECAD). Have you considered comparing ECMWF forecasts with SWG based on E-OBS, since both have an horizontal resolution of 0.25°x0.25° ?

⇒ **No, we have not used E-obs in this study. We did an extraction of the ECMWF forecast at the station level, we computed the CRPS and we did the comparison. We mention that we compared the values of CRPS (we found for ECMWF) to the CRPS provided by ECMWF for each month. It is true that E-obs have the same horizontal resolution as ECMWF forecast, but a resolution of 0.25°x 0.25° does not consider the information at the station level. As the E-OBS data are interpolations of ECA&D, both datasets are close near the four stations we consider (see table of correlation below). Moreover, we used E-obs to forecast precipitation near Madrid as an example, and the values of CRPSS/ correlation do not change that much. We will mention that in our results.**

| Station | Correlation between E-OBS and ECA&D (pearson) | Correlation between E-OBS and ECA&D (Spearman) |
|---------|---------|---------|
| Orly | 0.77 | 0.82 |
| Berlin | 0.69 | 0.77 |
| Madrid | 0.87 | 0.85 |
| Toulouse | 0.74 | 0.80 |

[Figure]

**Fig.Scatter plot of precipitation of Madrid from 1950 to 2020.**

- The evaluation of CRPSS conditional on weather regimes is interesting but I wonder whereas considering the weather regime of the last day of the sequence (t0+T) is representative for the weather regime of the whole sequence.
⇒ **Thank you for this interesting suggestion, as explained below we will consider this and we will change our methodology: we will consider the "dominating" weather regimes for each simulation (rather than at the last day) and re-verify this relation between CRPS and weather regimes.**

More minor, some references are missing (see below). There are issues in the units of CRPS. There are several equation issues.

Detailed comments :
- l 65 : Reference to Klein Tank is missing
- l 67 : please specify that ECAD provides point (station) data
- l 72-81 : actually ERA5 is now available since 1950.
- l 84 : Reference to Herschbach is missing. By the way, is it the right reference ?
- l 87 : so ECMWF forecasts have the same resolution as EOBS.
⇒ **We will take care of the references and those five comments**

- l 104 eq (1) : This is a good idea to account for several days in the distance, however D in (1) is not anymore a mathematical distance. Of course this is not mandatory for analog search, however why not using [sum_x sum_i {Z500(x,t+i)-Z500(x,t'+i)²} ], which is a mathematical distance? By the way, could you pelase provide a comparison of the results with the Euclidean distance (based on 1 day) vs. the D distance (based on 4 days)? And why 4 days ?
⇒ **We beg to disagree with the referee. The formula we use (square root of a sum of squares) is a distance in the mathematical sense, as it corresponds to a Euclidean distance in an augmented space. The formula given by the referee is NOT a distance because it does not satisfy the triangular inequality. An elementary counter example is to consider three distinct aligned points A, B, C, and B is between A and C. It should be clear to the reviewer that $d(A,C) > d(A,B) + d(B,C)$ when using the above formula (with no square root), because there an equality with sqrt(d). This contradicts the triangular inequality that is a property of mathematical distances.**
**We will show how the skill scores are improved with d=4, with respect to d=1. Yiou et al. (Clim. Dyn. 2013) argued for such delays for analog computations to simulate atmospheric fields. It is true that the distance based on 1 day is lower. However, 4 days help to better catch the persistence. That helps to obtain better skill scores for the forecast.**
**We prefer to use d<5 days, as we make forecasts for lead times T of 5 and 10 days.**

- l 111-118 : explanations are quite confusing. I had to read Yiou and Déandréis to understand Please consider rewritting the method.
⇒ **We will rewrite this part.**

- Figure 1 : a) please consider placing the red rectangle somewhere else within the 30 days for clarity since its date is not necessarily the same as the target day. b) the largest window doesn't match the coordinates given l 80. I would be happy to see some results on the other windows of analogy. Otherwise I think it's not worth showing them. Also there are several syntax issues in the caption

**⇒ We will modify the figure to make it clearer. For the results on the other windows of analogy, the small blue box is not different from the red rectangle. That is why we considered it. However there is a difference for the big blue rectangle, and we have put this information in table 2.**

- l 122 : please specify that persistence is computed over year k (unlike the climatology which is computed over all years)

**⇒ ok**

- l 124 : « control forecast » I dont't understand

**⇒ Indeed. It is not the appropriate word. We meant climatology and persistence forecasts. This will be changed.**

- l 137 : I guess that averaging the 100 trajectories smooths out the predictions. So at the end, is there a real gain (in terms of CRPSS) compared to considering only one analog ? (maybe that's already studied in another article, I haven't checked)

**⇒ The CRPS/CRPSS are based on the 100 trajectories, not by considering only the mean. The "averaged" trajectory is just for illustration purposes, and does not influence the CRPS score computations. The correlations are computed over the averaged trajectories.**

- l 150 : P(x) should be P(x,t) for day t. Please also rephrase the sentence

**⇒ OK.**

- l 153 eq (2) : the equation is confusing. Should be CRPS(P,t) and t should be in the right side as well. The inferiori limit is 0 for precipitation.

**⇒ we will verify and correct the equation to make it consistent.**

- l 159 : seasonality → climatology

**⇒ OK.**

- l 162 eq (3) : Equation issues. there should a sum over the days (or mean) in the numerator and denominator

**⇒ We will correct all this.**

- Table 1 : is it Pearson correlation ? I'm surprised that all pvalues equal 2.2^{-16} (I guess you mean 2.2 10^{-16}?).

**⇒ It is a Spearman (rank) correlation. We will use confidence intervals, which are more informative.**

- l 198-201 : this paragraph should go after l 204. Please refer to Fig 3.
⇒ **We will correct this.**

- l 212 : syntax issue
⇒ **we will correct this.**
- Table 2 : I guess this is for NCEP ?
⇒ **yes, indeed.**
- l 228 : remove the brackets
⇒ **OK.**

- l 230 : so you obtain 100 classifications. How do you deal with that ?
⇒ **We will clarify this paragraph. Indeed, we followed the procedure of Yiou et al. (NPG, 2008): we obtain 100 classifications. Then we classify the centroids of those 100 classifications and determine the most probable classification. This Monte-Carlo procedure helps "stabilizing" the classification into weather regimes.**

- l 241 : is the weather regime at time t0+T representative of the sequence from t0 to t0+T. Why don't you consider the most frequent WR within t0 to t0+T ?
⇒ **This is a very interesting suggestion. Actually we took the weather regime at t0+T and t0+T/2, which is not useful, as you point out. We decided to consider the most frequent weather regime in each simulation and determine how that will change the results.**

- Fig 3 : please consider plotting both reanalyses on the same plot for ease of comparison. Actually for persistence ERA5 seems to give larger CRPSS. Caption : persistence in lowercase letter. Please use either « reeference » or « baseline » along the article. For the boxplots, why don't you consider correlation with the mean instead of the median (as the predictions of SWG) ? (but it should not change much)
⇒ **We will add another plot containing both reanalyses, and we will take care of using reference in the whole paper.**

- l 243 : don't you mean below the 25th quantile ? Where is this used ?
⇒ **We noticed that all the values of CRPS below the 75th quantile are very low, so it doesn't matter to consider this quantile. This will be changed with the redefinition of the "most frequent weather regimes".**

- Figure 4 caption : blank space after (BLO)
⇒ **OK.**

- l 251 and followings : I see there are differences depending on the WR but it seems to depend on the city. Can we have explanations why, e.g. CRPS for BLO is better in Orly ?

**⇒ We related this to the difference in the local weather. However as we will make changes in the way of attributing CRPS to weather regimes we will see how that will change.**

- Figure 5 : Units of CRPS are not mm. What is the lead time here ? Given l 243-244, I would have expected here to see boxplots for the two classes of predictability.
**⇒ lead time of 5 days, that's what we are showing . This will be clarified. The units of CRPS are mm/day.**

- l 257 : what is the reference for CRPSS ? (I guess climatology)
**⇒ yes climatology. This will be clarified.**

- l 261 : do I understand correctly that in ECMWF forecasts, CRPSS is given for the whole of Europe whereas CRPS are avalaible at every grid point ? As said above, I find difficult comparing the skills of ECMWF vs SWG given that the horizontal resulution is different (0.25°x0.25° vs point data). Comparison of ECMWF with EOBS at the same resolution may be easier .
**⇒ Yes ECMWF forecasts are given for the whole of Europe, we extracted forecasts in single points which have the same coordinates then the studied stations, then we did the comparison. ECAD data are provided at station level (This is what interests us in this study). Moreover, the E-OBS are made from the ECAD data. By comparing the E-OBS and ECAD there is a strong correlation between data.**

- Table 3 : please specify the reference. You may want to add here the CRPSS of Europe with ECMWF.
**⇒ climatology. This will be clarified.**

- l 264 : CPRSS are actually hard to compare since they are not based on the same data (different resolution)
**⇒ We cannot find literature that explains how CRPS/CRPSS values should depend on data spatial resolution. Those scores are connected to the temporal resolution of variables and the size of ensembles. We will do the simulations with E-OBS data (that yields the same horizontal resolution as the ECMWF forecast) for comparison purposes. The main difference stems from the ensemble size.**

- l 266 « We found... » I don't understand the sentence (syntax issues). Anyway according to the CDF of CRPS in Fig 6, ECMWF seems significantly better (a much larger proportion of low values)
**⇒ We will correct the syntax issues to clarify this point.**

- l 270 again 2.2^{-16} ? Anyway, I think something's wrong here because the CDFs in Fig 6 do seem different. A difference of 0.2 between CDFs is large actually.
**⇒ small p-values of Kolmogorov Smirnov indicate that the null hypothesis is rejected, and it is in agreement with the D (difference between CDFs)**

- l 276 and Fig 7 : I think something's wrong because ECMWF shows a much larger proportion of small CRPS for Toulouse and Madrid (see Fig 6). The difference in CRPSS for Orly between 5 and 10 days is very surprising.

**⇒ We will compare what we find for precipitation with temperature in order to better understand this relation; for Toulouse and Madrid, we will verify this.**

- l 300 : designed
**⇒ ok**

- Some references are missing. There is no year for Cassou.
**⇒ We will take care of the references.**

---

## Author Comment (AC2)

**Review of the manuscript "Assessment of stochastic weather forecast of precipitation near European cities, based on analogs of circulation" by M. Krouma et al.**

⇒ **We thank the reviewer for the positive and constructive comments.**

This is a very interesting manuscript, owning a good potential to become a high impact paper with positive repercussions on different societal sectors. A stochastic rain generator is produced exploiting the relationship between Z500 and precipitation in different European cities. The work is worth publication, but it needs a substantial revision about three distinct points:

1 - an improved description of the methodology is needed, in order to better understand the workflow and some of the choices that have been employed.

⇒ **We will take care of this and rewrite the methodology in order to be more self contained.**

2 - the use of the prolonged ERA5 dataset (since 1950) is urged, in order to understand whether the differences in skill with NCEP are actually due to the length of the analog database, or to the database itself.

⇒ **The first reviewer also pleaded for this. We will extend the analogue search in ERA5 to 1950. It should be acknowledged that a few colleagues at ECMWF advised against such an extension due to an undocumented (yet) potential discontinuity before 1979.**

3 - a thorough and comprehensive revision of the English language is needed. Many subject-predicate inconsistencies, missing s', wrong sentence structures make some parts of the manuscript very hard to read.

⇒ **OK. We will be more careful with the English language in the revision.**

Specific comments are stored in the attached files. In bold font, those that pertain to the above-mentioned major observations.
Please also note the supplement to this comment: https://gmd.copernicus.org/preprints/gmd-2021-36/gmd-2021-36-RC2-supplement.pdf

Lines 79-80 and figure 1. Why exactly that region? Is there a process-knowledge approach behind this choice, a literature review or a regression/correlation between Z500 and rain over each station was applied? Whatever, the choice, it should be justified by means of references or, in the last case, with a graph/map that certifies the link between Z500 and precipitation.

⇒ **We justified our choice with references in the introduction. We mentioned some previous works where the relation of Z500 and precipitation was explained. We will add references to reflect this.**

Line 98. Why did the author choose exactly 20 analogues? Would not be better to base the choice on a maximum Euclidean distance?

⇒ **The choice of 20 analogues was based on experimental experience. First we considered 20 analogues to ensure that we have enough analogues dates for simulations,**

**second, we do not find changes on the Euclidean distance when we exceed some number of analogues. A theoretical study of Platzer et al. (2020, https://arxiv.org/pdf/2101.10640) shows that for complex systems the use of a large number of analogues (exceeding 30 analogues) doesn't make a big change for a forecast with analogs. Basing the results on a variable number of analogs (with a threshold distance value) would require complicated tests to choose an appropriate threshold, which might be season dependent.**

Line 100. The 4-day time embedding is not clear to me. Why is it necessary? Why does it preserve the temporal derivative of the atmospheric field? Please explain.

**⇒ A 4 day embedding enhances a better simulated persistence and yields better skill scores for the forecast. This was explained by Yiou et al. (Ensemble reconstruction of the atmospheric column from surface pressure using analogues. Clim. Dyn., 41, 1419-1437, 2013).**

Lines 113-118. Despite the mechanism is quite clear to me, the sentences "In order to go […] precipitation between t0 and t0 + T" are not well formulated. Since this is a crucial part for the understanding of the method, I would rephrase and expand this part.

**⇒ This will be better explained in the revision.**

Line 120. "of the properties" is redundant.

**⇒ OK.**

Line 123. More than the average value, the persistence consists in the anomaly between t0 - T and t0. Also, the climatological forecast takes.

Lines 128-134. Again, it is not clear upon which basis those domains (in Fig. 1b) have been chosen and the final domain selected among the four attempts. Also, the entire paragraph needs a language revision.

Line 150. A verb is missing (meet?).

**⇒ We will take care of those three comments. This will be rephrased and we will correct the mistakes.**

Figure 2. A visual legend is needed. Also, what's in the y-axis? Are the 5th and 95th percentiles calculated over the 1948-2019 time series?

**⇒ We will add a visual legend to make the figure clearer. The y-axis represents the mean of the precipitation computed over each lead time. And percentiles are computed over 1948 -2019.**

Line 184: six??

**⇒ We mean four (a mistake).**

Lines 201-205. It seems to me that part of the methodology is described here, where a description of results is expected.

**⇒ We will move this part to a new subsection on "parameter optimization".**

Line 206 and following. Very little description is given for figure 3. First of all, I think it is useful to illustrate all the stations in the main text, instead of showing only Orly while relegating the others to the Appendix. This is one of the main results of the study and deserves a better stage (instead, I would recommend to place Fig. 4 the Appendix, since the WRs are not a result of this study). Besides, fig. 3 and fig. A1 show a very interesting characteristic that should be discussed: most of the times, in fact, the summer SWG forecast vs. persistence improves with lead times, which is somewhat unexpected. Any thoughts about it?

⇒ **We thought that it is better to illustrate one station in the main text in order to not repeat the same information. Regarding the persistence, we find the same thing for other variables, and we think that could be related to the seasonality.**

Line 217. De Bilt?
Line 225. They help describe.
Line 232. Missing year for the Cassou reference.
⇒ **we will correct this and the missing reference.**

Lines 246-255. Fig. 5 (B1) should be described with more care. What do "Good forecasts (low quantiles of CRPS)" and "The low quality forecasts (high quantiles of CRPS)" mean? The caption for figure 5 is totally unclear and does not describe the plot. On the top of the blocking bar (panel b) a group of dots appear.
⇒ **We will redo the plot, what we meant here is that we attributed high and low quantiles of CRPS with respectively good and bad quality of the precipitation forecast as by definition the values of CRPS close to zero indicates a good forecast. The captions will be clarified.**

Lines 257-276. This paragraph is hard to read, there are many inconsistencies between subjects and predicates and other grammar errors. Also, the first paragraph is not clear: what is it meant to demonstrate? Maybe the lower skill of the ECMWF forecast? The latter is calculated over the entire European domain, how can it be compared with a forecast over single stations? A table with both ECMWF and SWG CRPSS would be more informative than a few words, if the authors find the way to make a fair comparison between the two.
⇒ **We will take care of the grammar errors and clarify this part by adding a table with CRPSS. We extracted the ECMWF forecasts at single points that have the same coordinates as the studied stations and we made the comparison.**

Line 279. The input of our model was analogs of geopotential heights at 500 hPa (Z500). This sentence should be rephrased.
⇒ **OK. we will correct this.**

Line 283. I cannot accept this conclusion. The only way to test it is to compute the analogs with the 70-year ERA5 dataset, now available since 1950. This is a very important test that should be included in this study, because it clarifies the role of the different reanalyses as

well as the role of database length.

**⇒ As we said before, we will compute analogs from the ERA5 dataset including data from 1950 to 1978. At a first glance, the results with ERA5 (1950-2019) and NCEP (1948-2019) are very similar.**

---

## Author Response (AR1)

**Review (1) of « Assessment of stochastic weather forecast of precipitation near European cities, based on analogs of circulation » by Krouma et al.**

**We thank the reviewer for the positive and constructive comments, which we considered in the new version of our manuscript.**

General comments :

The article assesses the skill of a stochastic weather generator to forecast precipitation in 4 cities of Western Europe. The SWG is based on random sampling of analogs of geopotential height. It was developed in another article by Yiou and Déandréis 2019, where it was applied to temperature. This study complements the latter for precipitation. As a refinement, a time embedding of 4 days is considered in the distance for the search of analogs (however the considered distance is not a mathematical distance anymore - see below). Skill scores are evaluated for lead times of 5 to 20 days. Results show positive skills up to 10 days. A comparison to ECMWF forecasts is provided but I have some concerns about this part (see below). The study is interesting, clear and well written. Precipitation forecasting is an important subject of research and I support the idea basing SWG on analogs. However I have several main concerns :

- The results are mainly shown for NCEP which used to cover a longer period than ERA5. However ERA5 is now available since 1950. Given its much better resolution, I recommend considering ERA5 for all the results.

⇒ **We extended the search of analogs to 1950 using the ERA5 database. We showed in section 4.1 that there is no difference between NCEP and ERA5 from 1950. And we decided to show results with NCEP in the paper.**

- I'm surprised that all the applied tests (Table 1 and p 14) have pvalues equal to $2.2^{-16}$ (I guess you mean $2.2 \cdot 10^{-16}$?). Isn't that strange ? More importantly, I doubt that Kolmogorov Smirnov gives such low pvalues given the differences in the CDF of Figure 6.

⇒ **For all the correlation tests we are showing in table 1,2,3 and 4 we computed the confidence interval. For the Kolmogorov Smirnov (K-S) test, the p-values were given by the K-S test. In fact the K-S test with low p-values meant the rejection of the null hypothesis that the two distributions are equal. Hence, the Figure 6 and the conclusion of the KS test are consistent and we explicitly said that.**

- I'm concerned about the comparison with ECMWF forecasts since ECMWF are gridded data, whereas SWG is based on point data (ECAD). Have you considered comparing ECMWF forecasts with SWG based on E-OBS, since both have an horizontal resolution of 0.25°x0.25° ?

⇒ **We used E-obs for clarifications and we found that they are giving the same results as ECA&D as explained before.**

- The evaluation of CRPSS conditional on weather regimes is interesting but I wonder whereas considering the weather regime of the last day of the sequence (t0+T) is representative for the weather regime of the whole sequence.

⇒ **We took into consideration this suggestion and we considered the most frequent weather regime for each simulation (rather than at the last day) as explained in**

**subsection 3.4 and we verified the relation between CRPS and weather regimes. We represent results in subsection 4.4**

More minor, some references are missing (see below). There are issues in the units of CRPS. There are several equation issues.

Detailed comments :
- l 65 : Reference to Klein Tank is missing
- l 67 : please specify that ECAD provides point (station) data
- l 72-81 : actually ERA5 is now available since 1950.
- l 84 : Reference to Herschbach is missing. By the way, is it the right reference ?
- l 87 : so ECMWF forecasts have the same resolution as EOBS.
**⇒ We added the missing references.**

- l 104 eq (1) : This is a good idea to account for several days in the distance, however D in (1) is not anymore a mathematical distance. Of course this is not mandatory for analog search, however why not using [sum_x sum_i {Z500(x,t+i)-Z500(x,t'+i)²} ], which is a mathematical distance? By the way, could you pelase provide a comparison of the results with the Euclidean distance (based on 1 day) vs. the D distance (based on 4 days)? And why 4 days ?
**⇒ We added the results of the simulation of the precipitation with analogs computed based on 1 day and 4 days embedding in the subsection 4.1. The comparison showed that the skill of the forecast with 4 days embedding is better than 1 day.**

- l 111-118 : explanations are quite confusing. I had to read Yiou and Déandréis to understand Please consider rewritting the method.
**⇒ We clarified the subsection 3.3.**

- Figure 1 : a) please consider placing the red rectangle somewhere else within the 30 days for clarity since its date is not necessarily the same as the target day. b) the largest window doesn't match the coordinates given l 80. I would be happy to see some results on the other windows of analogy. Otherwise I think it's not worth showing them. Also there are several syntax issues in the caption
**⇒ We modified the figure 1 a and b and made it clearer. We explained in subsection 4.1 the choice of the windows of analogy. In fact there is no difference between the small blue boxes and the red rectangle. However for the big blue rectangle we provided a table 1, where we are showing the difference between the two domains.**
- l 122 : please specify that persistence is computed over year k (unlike the climatology which is computed over all years)
**⇒ done**

- l 124 : « control forecast » I dont't understand
**⇒ We changed this.**

- l 137 : I guess that averaging the 100 trajectories smooths out the predictions. So at the end, is there a real gain (in terms of CRPSS) compared to considering only one analog ? (maybe that's already studied in another article, I haven't checked)

**⇒ The CRPS/CRPSS are based on the 100 trajectories, not by considering only the mean. The "averaged" trajectory is just for illustration purposes, and does not influence the CRPS score computations. The correlations are computed over the averaged trajectories.**

- l 150 : P(x) should be P(x,t) for day t. Please also rephrase the sentence

**⇒ We corrected the equation.**

- l 153 eq (2) : the equation is confusing. Should be CRPS(P,t) and t should be in the right side as well. The inferiori limit is 0 for precipitation.

**⇒ We corrected this equation as well.**

- l 159 : seasonality → climatology

**⇒ done.**

- l 162 eq (3) : Equation issues. there should a sum over the days (or mean) in the numerator and denominator

**⇒ We corrected it.**

- Table 1 : is it Pearson correlation ? I'm surprised that all pvalues equal $2.2^{-16}$ (I guess you mean $2.2 \cdot 10^{-16}$?).

**⇒ We are using a Spearman (rank) correlation. We used confidence intervals instead of p-values.**

- l 198-201 : this paragraph should go after l 204. Please refer to Fig 3.

**⇒ We corrected this.**

- l 212 : syntax issue

**⇒ done**

- Table 2 : I guess this is for NCEP ?

**⇒ done.**

- l 228 : remove the brackets

**⇒ done.**

- l 230 : so you obtain 100 classifications. How do you deal with that ?

**⇒ We clarified this part.**

- l 241 : is the weather regime at time t0+T representative of the sequence from t0 to t0+T. Why don't you consider the most frequent WR within t0 to t0+T ?

**⇒ We decided to consider the most frequent weather regime in each simulation and we get results that we discuss and present on subsection 4.4**

- Fig 3 : please consider plotting both reanalyses on the same plot for ease of comparison. Actually for persistence ERA5 seems to give larger CRPSS. Caption : persistence in lowercase letter. Please use either « reeference » or « baseline » along the article. For the boxplots, why don't you consider correlation with the mean instead of the median (as the predictions of SWG) ? (but it should not change much)

**⇒ We added a table 2 where we are shown the comparison between the simulations from both reanalyses.**

- l 243 : don't you mean below the 25th quantile ? Where is this used ?

**⇒ We considered values below the 25th quantile as indicator of good forecast quality, and quantile beyond 75th quantile for poor forecast quality**

- Figure 4 caption : blank space after (BLO)

**⇒ done.**

- l 251 and followings : I see there are differences depending on the WR but it seems to depend on the city. Can we have explanations why, e.g. CRPS for BLO is better in Orly ?

**⇒ We related this to the difference in the local weather.**

- Figure 5 : Units of CRPS are not mm. What is the lead time here ? Given l 243-244, I would have expected here to see boxplots for the two classes of predictability.

**⇒ That was clarified and the units of CRPS was corrected in the subsection 4.5.**

- l 257 : what is the reference for CRPSS ? (I guess climatology)

**⇒ yes climatology. This will be clarified.**

- l 261 : do I understand correctly that in ECMWF forecasts, CRPSS is given for the whole of Europe whereas CRPS are avalaible at every grid point ? As said above, I find difficult comparing the skills of ECMWF vs SWG given that the horizontal resulution is different (0.25°x0.25° vs point data). Comparison of ECMWF with EOBS at the same resolution may be easier .

**⇒ Yes ECMWF forecasts are given for the whole of Europe, we extracted forecasts in single points which have the same coordinates then the studied stations, then we did the comparison. ECAD data are provided at station level (This is what interests us in this study). Moreover, the E-OBS are made from the ECAD data. By comparing the E-OBS and ECAD there is a strong correlation between data.**

- Table 3 : please specify the reference. You may want to add here the CRPSS of Europe with ECMWF.

**⇒ climatology. This was clarified.**

- l 264 : CPRSS are actually hard to compare since they are not based on the same data (different resolution)

**⇒ We cannot find literature that explains how CRPS/CRPSS values should depend on data spatial resolution. The main difference stems from the ensemble size. We did the simulations with E-OBS data (that yields the same horizontal resolution as the ECMWF forecast) and we found the same results.**

- l 266 « We found... » I don't understand the sentence (syntax issues). Anyway according to the CDF of CRPS in Fig 6, ECMWF seems significantly better (a much larger proportion of low values)
**⇒ We corrected this.**

- l 270 again 2.2^{-16} ? Anyway, I think something's wrong here because the CDFs in Fig 6 do seem different. A difference of 0.2 between CDFs is large actually.
**⇒ small p-values of Kolmogorov Smirnov indicate that the null hypothesis is rejected, and it is in agreement with the D (difference between CDFs) we explained this in subsection 4.5.**

- l 276 and Fig 7 : I think something's wrong because ECMWF shows a much larger proportion of small CRPS for Toulouse and Madrid (see Fig 6). The difference in CRPSS for Orly between 5 and 10 days is very surprising.
**⇒ We will compare what we find for precipitation with temperature in order to better understand this relation; for Toulouse and Madrid, we will verify this.**

- l 300 : designed
**⇒ done.**

- Some references are missing. There is no year for Cassou.
**⇒ We added all the references.**

**Review (2) of the manuscript "Assessment of stochastic weather forecast of precipitation near European cities, based on analogs of circulation" by M. Krouma et al.**

**We thank the reviewer for the positive and constructive comments, which we considered in the new version of our manuscript.**

This is a very interesting manuscript, owning a good potential to become a high impact paper with positive repercussions on different societal sectors. A stochastic rain generator is produced exploiting the relationship between Z500 and precipitation in different European cities. The work is worth publication, but it needs a substantial revision about three distinct points:

1 - an improved description of the methodology is needed, in order to better understand the workflow and some of the choices that have been employed.

⇒ **We rewrote the methodology, we clarified our choices for the analog search and the configuration of the SWG. ( subsection 3.1 & 3.2)**

2 - the use of the prolonged ERA5 dataset (since 1950) is urged, in order to understand whether the differences in skill with NCEP are actually due to the length of the analog database, or to the database itself.

⇒ **We extended the analog search in ERA5 to 1950. And we did a comparison with NCEP. We represented the results on a table 2 subsection 4.1.**

3 - a thorough and comprehensive revision of the English language is needed. Many subject-predicate inconsistencies, missing s', wrong sentence structures make some parts of the manuscript very hard to read.

⇒ **We took care of the English language in this new version of the paper.**

Specific comments are stored in the attached files. In bold font, those that pertain to the above-mentioned major observations.

Please also note the supplement to this comment:
https://gmd.copernicus.org/preprints/gmd-2021-36/gmd-2021-36-RC2-supplement.pdf

Lines 79-80 and figure 1. Why exactly that region? Is there a process-knowledge approach behind this choice, a literature review or a regression/correlation between Z500 and rain over each station was applied? Whatever, the choice, it should be justified by means of references or, in the last case, with a graph/map that certifies the link between Z500 and precipitation.

⇒ **We justified our choice with references in the introduction, we mentioned recent studies where the relation of Z500 and precipitation was explained. Regarding the geographical region, we explained further in subsection 4.1, that we explored the relationship between different regions. However, we found that the region with the correlations 30°W-20°E; 40°-60°N is the optimal as it allows to calculate analogs for the different regions and also at a lower cost.**

Line 98. Why did the author choose exactly 20 analogues? Would not be better to base the choice on a maximum Euclidean distance?

**⇒ We explained that the choice of 20 analogs was based on experimental experience. We explained that we do not find changes on the Euclidean distance when we exceed some number of analogs. We justified our choice using a recent study of Platzer et al. (2020, https://arxiv.org/pdf/2101.10640) that shows that for complex systems the use of a large number of analogs (exceeding 30 analogues) doesn't make a big change for a forecast with analogs.**

Line 100. The 4-day time embedding is not clear to me. Why is it necessary? Why does it preserve the temporal derivative of the atmospheric field? Please explain.

**⇒ We explained that 4 day embedding enhances a better simulated persistence and yields better skill scores for the forecast. We added the results of that on subsection 4.1.**

Lines 113-118. Despite the mechanism is quite clear to me, the sentences "In order to go […] precipitation between t0 and t0 + T" are not well formulated. Since this is a crucial part for the understanding of the method, I would rephrase and expand this part.

**⇒ We rephrased the subsection 3.3.**

Line 120. "of the properties" is redundant.

**⇒ We corrected this.**

Line 123. More than the average value, the persistence consists in the anomaly between t0 - T and t0. Also, the climatological forecast takes.

Lines 128-134. Again, it is not clear upon which basis those domains (in Fig. 1b) have been chosen and the final domain selected among the four attempts. Also, the entire paragraph needs a language revision.

Line 150. A verb is missing (meet?).

**⇒ We rephrased and we corrected the mistakes.**

Figure 2. A visual legend is needed. Also, what's in the y-axis? Are the 5th and 95th percentiles calculated over the 1948-2019 time series?

**⇒ We added a visual legend to make the figure clearer. And we clarified the caption of figure 3.**

Line 184: six??

**⇒ We meant four (a mistake).**

Lines 201-205. It seems to me that part of the methodology is described here, where a description of results is expected.

**⇒ We added this result on a new subsection 4.1 "parameter optimization", where we showed the results of the different experiments that we did (databases, geographical domain, embedding..)**

Line 206 and following. Very little description is given for figure 3. First of all, I think it is useful to illustrate all the stations in the main text, instead of showing only Orly while relegating the others to the Appendix. This is one of the main results of the study and deserves a better stage (instead, I would recommend to place Fig. 4 the Appendix, since the WRs are not a result of this study). Besides, fig. 3 and fig. A1 show a very interesting characteristic that should be discussed: most of the times, in fact, the summer SWG forecast vs. persistence improves with lead times, which is somewhat unexpected. Any thoughts about it?

**⇒ We illustrated the results of the different studied stations. We chose to show results with NCEP to be more precise.**

Line 217. De Bilt?
Line 225. They help describe.
Line 232. Missing year for the Cassou reference.

**⇒ we corrected the sentence and added the missing reference.**

Lines 246-255. Fig. 5 (B1) should be described with more care. What do "Good forecasts (low quantiles of CRPS)" and "The low quality forecasts (high quantiles of CRPS)" mean? The caption for figure 5 is totally unclear and does not describe the plot. On the top of the blocking bar (panel b) a group of dots appear.

**⇒ We clarified the caption in figure 6, we explained further the plot, and how we sampled the weather regimes dates. We explained on the subsection 3.4 how we determined the relation with the CRPS, our definition of good/low forecast quality.**

Lines 257-276. This paragraph is hard to read, there are many inconsistencies between subjects and predicates and other grammar errors. Also, the first paragraph is not clear: what is it meant to demonstrate? Maybe the lower skill of the ECMWF forecast? The latter is calculated over the entire European domain, how can it be compared with a forecast over single stations? A table with both ECMWF and SWG CRPSS would be more informative than a few words, if the authors find the way to make a fair comparison between the two.

**⇒We rephrase the paragraph. And we explained that we extracted the ECMWF forecasts at single points that have the same coordinates as the studied stations and we made the comparison.**

Line 279. The input of our model was analogs of geopotential heights at 500 hPa (Z500). This sentence should be rephrased.

**⇒ We corrected the sentence.**

Line 283. I cannot accept this conclusion. The only way to test it is to compute the analogs with the 70-year ERA5 dataset, now available since 1950. This is a very important test that should be included in this study, because it clarifies the role of the different reanalyses as well as the role of database length.

**⇒ We computed analogs from the ERA5 dataset including data from 1950 to 1978. The results with ERA5 (1950-2019) and NCEP (1948-2019) are very similar, we are showing this in the subsection 4.1.**

---

## Author Response (AR2)

**We thank the reviewers for the constructive comments, which we considered in the new version of our manuscript.**

—-------------------------------

**Report 1**

General comments :

The authors have made a good job in accounting for the reviewers' comments. The article is clearer and easier to read. My only significant point is the comparison with ECMWF and in particular what regards Fig 8 that seems surprising to me given the previous results of Section 4.5 (see below).

Detailed comments :

- abstract : « Results show significant positive skill score » : do you refer here to the comparison with ECMWF or with the persistence/climatology ?

⇒ **we refer to the comparison with persistence and climatology. This is clarified in the manuscript.**

- l 100 : brackets

⇒ **ok**

- l 136 : SGW

⇒ **ok, changed to SWG.**

- l 151 : brackets

⇒ **ok, we verified the reference formatting.**

- eq (2) is still wrong. You have P and xa on the left hand side while you have P_{a,t} and xa on the right hand-side. Both sides should match.

⇒ **The equation is corrected to be consistent.**

$$CRPS(P, x_a) = \int_{-\infty}^{+\infty} (P(x) - H(x - x_a))^2 \, dx$$

**where x is the forecast, $x_a$ is the observation, P is the cumulative distribution function of x, H is the Heaviside function of the occurrence of $x_a$ (H(z) = 0 when z<0, and H(z)=1 when z>= 0).**

- eq (3) : please specify that bar is the mean

⇒ **ok**

- l 173 : brackets

⇒ **ok**

- Table 2 : why do you consider two periods for ERA5 ? ERA5 extended might be enough.
In the caption : not only winter.

⇒ **At the time of writing, Dr. Florian Pappenberger (from the ECMWF) warned us that the extension of ERA5 prior to 1979 was "experimental" as it was obtained with different constraints than for 1979-now. In practice, the Copernicus Data Store also indicates that ERA5 data from 1950 and 1978 is experimental. In that respect, it made sense to conservatively keep the two periods of ERA5. For simplicity, we now only consider the extended ERA5 product (1950-now).**
**The caption is corrected.**

- l 245 brackets

⇒ **ok**

- l 272 « albeit significant » : what did you test exactly ?

⇒ **This was reformulated.**

- l 285 : brackets

**⇒ ok**

- l 296-297 : « We found that the values of CRPS of ECMWF forecast and SWG forecast are 80%, 39% 50% and 40 % equal or near to zero for respectively Orly, Berlin, Madrid and Toulouse » : there are two forecasts and 4 cities but you give only five % so I don't understand what they correspond to.

**⇒ This has been clarified in the text: for lead times of 5 days, CRPS values are very small XX% of the time.**

Also do you mean CRPSS ?

**⇒ We compared CRPS not CRPSS. This will be clarified and emphasized in the text.**

- l 291 « They are about 0.16 in the summer (JJA) and 0.25 in the winter (DJF) for a lead time of T = 5 days » : are these numbers right ? This seems quite bad (=low) for CRPSS. I'd have expected better CRPSS given Fig 7.

**⇒ Those are the values of ECMWF not ours; they are not related to Fig 7.**

**The values of the CRPSS come from the ECMWF forecast for summer (June July August - JJA) and winter (December January February - DJF) for Europe (Lat 35.0 to 75.0, lon -12.5 to 42.5) for 2019 and 2020. We read the values from the following figures (taken from the ECMWF** website **(https://apps.ecmwf.int/webapps/opencharts/products/plwww_3m_ens_tigge_wp_mean?area=Europe¶meter=24h%20precipitation&score=CRPSS) and the technical report from ECMWF (Haiden et al, 2020). Those two figures below show the rather large range of skill score values across models. We used the ECMWF values to qualitatively compare the skill of the SWG forecast. We mentioned that those values are for a large region (whole Europe). However, the aim is to say that we are consistently close to those values, as we are making forecasts for local sites. This was clarified in the text.**

[Figure]

**season:JJA**

total precipitation
Continuous ranked probability skill score
Europe (lat 35.0 to 75.0, lon -12.5 to 42.5)

| | 2019 KMA | 2020 KMA |
| | 2019 CMC | 2020 CMC |
| | 2019 JMA | 2020 JMA |
| | 2019 NCEP | 2020 NCEP |
| | 2019 UKMO | 2020 UKMO |
| | 2019 ECMWF | 2020 ECMWF |

[Figure]

[Figure]

**season:DJF**

total precipitation
Continuous ranked probability skill score
Europe (lat 35.0 to 75.0, lon -12.5 to 42.5)

| | 2019 KMA | 2020 KMA |
| | 2019 CMC | 2020 CMC |
| | 2019 JMA | 2020 JMA |
| | 2019 NCEP | 2020 NCEP |
| | 2019 UKMO | 2020 UKMO |
| | 2019 ECMWF | 2020 ECMWF |

[Figure]

- l 293 : « The CRPSS of SWG for a lead time of T = 5 days showed in Table 2, and this suggests.. » : syntax issue

**=> This is corrected.**

- Fig 7 : The unit of CRPSS is not mm

**⇒ it is a CRPS not a CRPSS. By definition, CRPS has the same units as precipitation (here mm/day).**

- Fig 8 : I'm sorry but I still don't understand these results (see my previous review). If I understood correctly, CRPSS here is 1-mean(CRPS ECMWF)/mean(CRPS SWG) so a CRPSS value of 0.5 means that mean(CRPS ECMWF)=1.5*mean(CRPS SWG). Given Fig 7, it's hard to believe that for Orly mean(CRPS ECMWF)=1.5*mean(CRPS SWG). From my eyes, they are almost equal so I'd have expected a CRPSS value for T=5 around 0. For Madrid, it seems to me that the average CRPS is much lower for ECMWF than for SWG so again I'd not have expected negative CRPSS.

Maybe my eyes are confused. If so it could be useful to add a line showing the means in **Fig 7**. Another possibility would be **to replace Fig 7 by boxplots**.

**⇒ This is indeed an intriguing point. The CDFs of CRPS values show a lower median for the ECMWF than SWG simulations. This is what a visual inspection shows. Yet, the mean CRPS is higher for ECMWF than for SWG simulations, because of outlier values of CRPS for ECMWF. CRPSS compares the means, not the medians. This will be illustrated with an improved Figure 7.**

For the case of Orly, I'm also still surprised (see my previous review) about the strong change in CRPSS between 5-10-20 days. **In your next response, I'd be happy to see the equivalent of Fig 7 for T=10 and 20**.

**⇒ ok**

**For clarification, we do the comparison based on the CRPS values of the SWG forecast and the ECMWF forecast. Then we compute the CRPSS between the mean CRPS of the ECMWF forecast and the mean CRPS of the SWG forecast.**

**In the following table (Table 1) , we compare the CRPSS values (computed between the mean of CRPS of ECMWF forecast and mean CRPS of SWG forecast). We considered the SWG forecast as a reference for the ECMWF forecast.**

**For a lead time of 5 days, as shown in the following table, we find that:**

**the CRPSS values are different from one station to another. We find a negative value of CRPSS for Orly, Berlin and Madrid, which means that the SWG has a skill compared to ECMWF forecast, which is not the case for Toulouse, where the ECMWF forecast is still significant. To explain those results, we took a look at the boxplots as suggested and the means of the CRPS of two forecasts (Table 2).**

**Table 1. CRPSS between forecast of ECMWF and SWG forecast for T = 5, 10, 20 days.**

|  | **Orly** | **Berlin** | **Madrid** | **Toulouse** |
|---|---|---|---|---|
| **CRPSS T = 5 days** | -0.09 | -0.02 | -0.2 | 0.25 |

| | | | | |
|---|---|---|---|---|
| **CRPSS T = 10 days** | -0.17 | -0.54 | -0.33 | 0.23 |
| **CRPSS T = 20 days** | -0.50 | -0.36 | -0.1 | -0.08 |

- **CDF for lead time of 5 days:**

[Figure]

[Figure]

[Figure]

- **CDFs for lead time of 5, 10, 20 days for Orly ( Example):**

[Figure]

- **Boxplots between CRPS of ECMWF and SWG taking Orly as example (outliers above 5mm/day do not appear on the boxplots):**

[Figure]

[Figure]

[Figure]

**Table 2. CRPSS and average of ECMWF and SWG forecasts for T = 5, 10, 20 days.**

|  | Orly | Berlin | Madrid | Toulouse |
|---|---|---|---|---|
| $\overline{CRPS}_{ECMWF}$ / Median | **1.87** / 0.04 | **16.56** / 0.05 | **18.73** / 0.003 | **12.76** / 0.01 |
| $\overline{CRPS}_{SWG}$ / Median | **1.70** / 0.67 | **16.10** / 10.37 | **15.49** / 5.45 | **17.16** / 8.39 |
| CRPSS 5 days | **-0.09** | **-0.02** | **-0.2** | **0.25** |
| $\overline{CRPS}_{ECMWF}$ | 1.70 | 18.1 | 20.03 | 14.87 |
| $\overline{CRPS}_{SWG}$ | 1.44 | 11.67 | 15.04 | 19.45 |
| CRPSS 10 days | **-0.17** | **-0.54** | **-0.33** | **0.23** |
| $\overline{CRPS}_{ECMWF}$ | 1.67 | 13.54 | 17.89 | 17.8 |
| $\overline{CRPS}_{SWG}$ | 1.11 | 9.91 | 16.23 | 16.41 |

| CRPSS 20 days | -0.50 | -0.36 | -0.1 | -0.08 |
|---|---|---|---|---|

Comparing the average of the CRPS of ECMWF and CRPS of SWG, we find that they are close, however the average of the CRPS of SWG is still smaller than the one from ECMWF, that could be explained by the fact that the CRPS of ECMWF contains more outliers as shown in the boxplots. This explains why we have those values of CRPSS. However, ECMWF has the smallest median (as shown in the boxplots) CRPS.

**#Report 2**

Despite the revised version has improved some aspects of the manuscript, e.g. throughout the use of ERA5 from 1950 for comparison to NCEP, the methodological choices are still poorly explained. In my opinion, the work still needs to address a few important points before being accepted for publication in GMD.

In particular, I am still concerned about the choice of the analogs domain:

- First of all, it is not clear why this information is in the Results section, and not in the methods. I see there was a Paragraph 3.3 (erased), which is probably the right place where this details should be expanded.

⇒ **We re-organized the manuscript by adding a new section "optimization of the parameter of the forecast" between the methodology and results. In this new section 4, we put all the details (choose of analogs, domain, the comparison between ERA5 and NCEP).**

- Figure 1 is not useful to answer this question, since it only shows four boxes where the sensitivity to the large scale was tested.

=> **The re-organization of the manuscript should solve this issue.**

- Table 1 compares two of these boxes in terms of correlation, **to indicate that the chosen box is better than the biggest of the remaining three boxes.**

⇒ **results from the small box are the same as the red one. This is explained in the text.**

- What I'd like to see, is a regression map of Z500 for rainfall in each of the four cities. This map will finally show the relationship between gridpoint precipitation and Z500 in the Euro-Atlantic region. **This map could be done for one season, while the other season may be shown in the supp. mat.**

⇒ **ok**

**Here we show the maps of rank correlation between the daily average of Z500 over the Euro Atlantic region and the precipitation in each studied station in order (Madrid, Berlin,Toulouse and Orly). We did the analysis for different seasons also and used a threshold for precipitation (> 1 mm/day). We find a maximum correlation amplitude of -0.5 for Madrid and Orly. We find a correlation of -0.4 and -0.3 for Toulouse and Berlin, respectively. The correlation is significant as we have a p.value < 0.05 for the different grid points. This indicates the relation between Z500 patterns and precipitation especially in western Europe and that a decrease in Z500 is linked with precipitation.**

[Figure]

[Figure]

- Also, the fact that two seasons are analyzed in this work is not at all clear since the beginning. JJA and DJF come out of the blue in tables 1 and 2.

⇒ **The choice of the two "extreme" seasons will be explained in the text. One pragmatic reason is to emphasize the seasonal dependence, which is moderate for the two extreme seasons. This leads to rather redundant figures/results with intermediate seasons.**

- I appreciate the effort of explaining the choice of the 20 analogs, since a larger amount does not improve the results, but I think this description is incomplete. What about less analogs (5, 10, 12...)? Also, once we have 20 dates, are precipitation averaged over these days in order to obtain rainfall prediction? This should be explained.

**⇒ ok (we added this result in the new section "SWG parameter optimization")**
**In the table below we are showing the score for simulations of the precipitation with SWG with different numbers of analogs 5 and 10. We notice that the scores (correlation and CRPSS) increase by increasing the number of analogs, which could be explained by raising the number of selection of analog dates. That justify the use of 20 analogs in this study. After selection of 20 dates, the predicted precipitation is the average of the precipitation on those analog dates (as we explained in section 3.2)**

| | | Orly | Berlin | Madrid | Toulouse |
|---|---|---|---|---|---|
| **Scores for simulations with 5 analog** | **Correlation** | **0.20** | **0.29** | **0.34** | **0.12** |
| | **CRPSS/ Persistence** | **0.34** | **0.29** | **0.32** | **0.34** |
| | **CRPSS/ Climatology** | **0.12** | **0.20** | **0.31** | **0.24** |
| **Scores for simulations with 10 analog** | **Correlation** | **0.20** | **0.29** | **0.38** | **0.13** |
| | **CRPSS/ Persistence** | **0.40** | **0.39** | **0.40** | **0.52** |
| | **CRPSS/ Climatology** | **0.23** | **0.31** | **0.39** | **0.45** |

- Section 3.4. The detailed description of weather regimes is not needed here, nor it is figure 2. In my understanding, WR are only needed to explain the forecast skill dependence on them, **therefore this section should ONLY describe how the authors assess the influence of WRs on the forecast quality.**
**⇒ We added this section, mainly to explain how we evaluated the relation between WRs and the forecast quality.**

Finally, as I reported in my first review, I would like to see some **more discussion on figure 4**. This plot is key and it needs better descriptions, some discussion (for example about why the forecast vs persistence improves/deteriorates with time, with no clear pattern) and more accurateness (e.g. line 294, "the CRPSS for persistence" should be "the CRPSS against the persistence reference").
**⇒ ok, we developed and explained further figure 4 (subsection 5.2). We linked the improvement / deterioration of the CRPSS with time to the number of rainy days in**

summer and winter. Indeed, we notice that the values of CRPSS against persistence reference (represented by squares) decrease with lead times in winter for the different studied areas, showing high values over 5 days. However, for summer, we notice that the values of CRPSS versus persistence increase with lead time, with high values over 20 days except for Berlin. This indicates that for the summer until 20 days the SWG forecast is still better than the persistence forecast (the average of the CRPS of SWG is smaller than the average of the CRPS of the persistence).

We computed the seasonal frequency of precipitation (defined as the number of days when precipitation exceeds 0.5 mm/day: table below). Precipitation exceeding 0.5 mm/day is more frequent in Berlin than in the other stations (close to 50% of the time for both seasons). This means that a persistence forecast is likely to be skillful, even for longer lead times, especially in the summer. Summer precipitation in Orly comes in cluster. Therefore, the trends in CRPSS values for different lead times are probably due to the intrinsic time persistence of local precipitation.

|  | Orly | Berlin | Toulouse | Madrid |
|---|---|---|---|---|
| % Rainy days summer | 51 | 53 | 45 | 31 |
| % Rainy days winter | 38 | 45 | 31 | 13 |

---

## Author Response (AR3)

**We thank the reviewer for reading our manuscript carefully and for the comments. This document is a point-by-point reply to the comments.**

Report 2
I acknowledge that the authors improved the manuscript much, following my suggestions, and I am thankful for that.

Unfortunately, there are still imprecisions, therefore the work cannot be published in the present form. I list a few of them, but authors are urged to carefully go through the text to spot more:
- Figure 2 caption does not correspond to the text description: the caption assigns Zonal Flow to summer and Atlantic Low to winter, the text says the opposite.
**We corrected the caption.**
- Figure 2 caption itself is not clear. Atlantic Zonal (NAO+) does not mean anything.
**This is corrected.**
- Figure 4: fonts are not homogeneous throughout panels.
**We edited figure 4.**
- Figure 5-6-A1 introduce the concept of Zonal Low, not described in the text. Is it the Atlantic Low?
**This is corrected.**
- Section 4: I would not use the acronym SWG in the section's title, authors should use Stochastic Weather Generator
**That has been changed in the text.**
- Conclusions: Lines 392-395 "We found that the NCEP and ERA5 extended reanalyses provide good performances for simulations, due to its longer length (≈ 70 years in NCEP and ERA5). Therefore the length of the analog database does make a difference, as already suggested by Jézéquel et al. (2018a)." This phrase makes no sense after performing the analysis with the extended ERA5 dataset.
**We deleted this sentence.**

Finally, many sentences are still convoluted and there are grammatical errors here and there. The authors are suggested to carefully revise the text with the help of a native English speaker.
**We took care of English grammar errors (we also understand that the text will be copy-edited by Copernicus before publication) and we rephrased some sentences to make them more linear.**